# KnowHalu: Hallucination Detection via Multi-Form Knowledge Based Factual Checking

## Abstract

As large language models (LLMs) become increasingly integral to a wide array of applications, ensuring the factual accuracy of their outputs and mitigating hallucinations is paramount. Current approaches, which primarily rely on *self-consistency checks* or *post-hoc fact-checking*, often fall short by disregarding the nuanced structure of queries and the diverse forms of contextual knowledge required for accurate response generation. To address these shortcomings, we introduce KnowHalu (pronounced "No Halu"), the first multi-form knowledge-based hallucination detection framework. We also introduce a new category of hallucinations, off-target hallucinations, which occur when responses are factually accurate but irrelevant or nonspecific to the query (e.g., answering "What's the primary language in Barcelona?" with "European language"). In particular, KnowHalu employs a rigorous two-phase process to detect hallucinations. In the first phase, it isolates off-target hallucinations by analyzing the semantic alignment between the response and the query. In the second phase, it conducts a novel multi-form knowledge-based fact-checking through a comprehensive pipeline of reasoning and query decomposition, knowledge retrieval, knowledge form optimization, judgment generation, and judgment aggregation. Extensive evaluations demonstrate that KnowHalu significantly surpasses state-of-the-art (SOTA) baselines across diverse tasks, achieving over 15% improvement in question answering (QA) and 6% in summarization tasks when applied to the same underlying LLM. These results underscore the effectiveness and versatility of KnowHalu, setting a new benchmark for hallucination detection and paving the way for safer and more reliable LLM applications.

## 1 Introduction

Significant advancements have been achieved in the field of Natural Language Processing (NLP) with the advent of Large Language Models (LLMs). While these models excel in generating coherent and contextually relevant text, they are prone to '*hallucinations*' — generating plausible but factually incorrect or unspecific information (Bang et al., 2023). This poses a considerable challenge, especially in applications demanding high factual accuracy, such as medical records analysis (Singhal et al., 2023), finance (Wu et al., 2023; Yang et al., 2023), and drug design (Vert, 2023; Savage, 2023).

To mitigate or detect hallucinations in LLMs, a series of approaches have been explored. For instance, self-consistency-based approaches detect hallucinations by identifying contradictions in responses that are stochastically sampled from the LLMs in response to the same query (Wang et al., 2022; Manakul et al., 2023; Mündler et al., 2023). Other approaches detect hallucinations by probing LLMs' hidden states (Azaria & Mitchell, 2023) or output probability distributions (Manakul et al., 2023). These methods do not incorporate external knowledge and are thus limited by LLMs' internal knowledge. Post-hoc fact-checking approaches have been recently shown to be effective even when LLMs' internal knowledge proves inadequate and achieved SOTA hallucination detection (Peng et al., 2023; Semnani et al., 2023). However, due to the limitation of LLM reasoning capabilities, even if the extracted knowledge is correct, the models may still struggle to perform factual checking accurately, especially with complex queries or logic, such as multi-hop queries or those involving multiple factual assertions. Thus, how to fully leverage the inherent reasoning capabilities of the model is important.

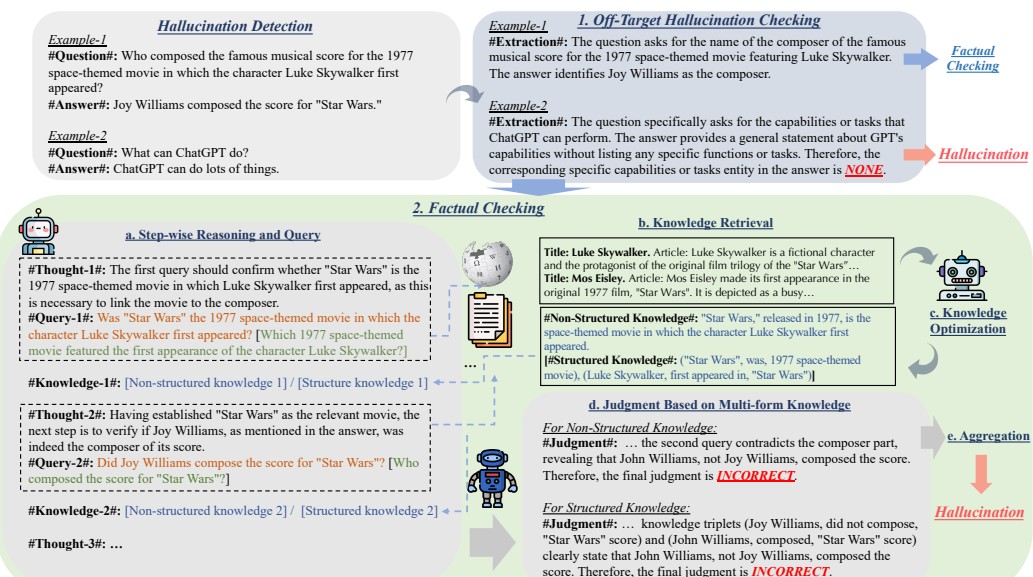

Figure 1: Overview of KnowHalu. The hallucination detection process starts with "*Off-Target Hallucination Checking*", focusing on the early identification of off-target hallucinations by scrutinizing the specificity of the answers. For potential fabrication hallucinations, KnowHalu then provides a comprehensive "*Factual Checking*", which consists of five steps: (a) "*Step-wise Reasoning and Query*" breaks down the original query into step-wise reasoning and sub-queries for detailed factual checking; (b) "*Knowledge Retrieval*" retrieves unstructured knowledge via RAG and structured knowledge in the form of triplets for each sub-query; (c) "*Knowledge Optimization*" leverages LLMs to summarize and refine the retrieved knowledge into different forms; (d) "*Judgment Based on Multi-form Knowledge*" employs LLMs to critically assesses the answer to sub-queries, based on each form of knowledge; (e) "*Aggregation*" provides a further refined judgment by aggregating predictions based on different forms of knowledge.

Recognizing this gap, our work proposes a novel multi-phase hallucination detection mechanism, KnowHalu (pronounced as "No halu"), the overall framework of which is presented in Figure 1. In particular, we first perform the *off-target hallucination checking*, where the answer indeed provides a fact but it is not a helpful response to the question, such as the answer "*ChatGPT can do lots of things*" for the question "*What can ChatGPT do?*" This type of hallucination has been extensively discussed in various works on hallucination in LLMs (Huang et al., 2023; Zhang et al., 2023b; Li et al., 2023). Despite this, the step of identifying off-target hallucinations remains critically underrepresented in current detection methodologies. Existing approaches often fail to discern answers that are factually correct but irrelevant to the posed questions. We then perform a *step-wise decomposition of queries*, which enables targeted retrieval of external knowledge pertinent to each logical step. For each decomposed logical step, we will perform the *multi-form knowledge based factual checking*, leveraging both the unstructured knowledge (e.g., normal semantic sentences) and structured knowledge (e.g., object-predicate-object triplets). This multi-form knowledge analysis captures a comprehensive spectrum of factual information, enhancing the reasoning capability of LLMs and ensuring a robust and thorough validation of each piece of retrieved knowledge. Finally, we perform the reasoning step by composing the step-wise factual checking results together and guide the LLMs to make the final judgment by providing related demonstrations. Our multi-step and multi-form knowledge based factual checking not only improves the verification accuracy but also enhances the model's ability to handle intricate and layered queries.

Our extensive experimental evaluations reveal that KnowHalu significantly outperforms state-of-the-art (SOTA) baselines in detecting hallucinations. The experiments, conducted across diverse datasets and tasks, demonstrate not only the high accuracy of our method in factual verification but also its versatility in handling various types of queries.

In summary, we make the following key contributions:

- We introduce KnowHalu, a novel approach with two main phases (off-target hallucination detection and multi-step factual checking) for detecting hallucinations in texts generated by LLMs, leveraging multi-form knowledge for factual checking. In particular, we define the categories of *off-target hallucinations* for the first time.

Table 1: Different Types of Hallucinations for Question-Answer (QA) Task. (Row 1-5 showcase off-target hallucinations, and the last row shows fabrication hallucinations.)

| Type of Hallucination | Category | Description | Example |
|---|---|---|---|
| Off-Target Hallucination | Vague or Broad Answers | Answers that are too general and do not address the specificities of the question. | **#Question#**: What is the primary language in Barcelona? **#Answer#**: European languages. |
| | Parroting or Reiteration | The response simply echoes part of the question without adding new or relevant information. | **#Question#**: What is the title of John Steinbeck's novel about the Dust Bowl? **#Answer#**: Steinbeck wrote about the Dust Bowl. |
| | Misinterpretation of Question | Misunderstanding the question, leading to an off-topic or irrelevant response. | **#Question#**: What is the capital of France? **#Answer#**: France is in Europe. |
| | Negation or Incomplete Information | Pointing out what is not true without providing correct information. | **#Question#**: Who is the author of "Pride and Prejudice"? **#Answer#**: Not written by Charles Dickens. |
| | Overgeneralization or Simplification | Overgeneralizing or simplifying the answer. | **#Question#**: What types of movies has Christopher Nolan worked on? **#Answer#**: Biographical film. |
| Factual Hallucination | Fabrication | Introducing false details or assumptions not supported by the truth of facts | **#Question#**: When was "The Sound of Silence" released? **#Answer#**: 1966 *(Incorrect. The correct answer is 1964)* |

- We are the first work to explore the influence of both the formulations of the queries and the forms of knowledge used for detecting hallucinations, highlighting our novel exploration into factors critical for improving hallucination detection accuracy.

- We further propose a verification mechanism where collections of facts are checked interdependently instead of in parallel, and an aggregation methodology based on the prediction results from different forms of knowledge to further reduce hallucinations in judgment itself. Our experiments show that our method achieves at least a **15.50%** improvement in hallucination detection on the question-answering task, and yields at least an additional **6.20%** improvement in the text summarization task when compared to the SOTA baselines using the same underlying LLMs.

## 2 RELATED WORK

**Hallucination of LLMs.** Hallucination in the LLM literature generally refers to LLMs generating nonfactual, irrelevant, or unspecific outputs. Such phenomena have been observed in a variety of tasks (Huang et al., 2023; Zhang et al., 2023b; Ji et al., 2023), such as translation (Lee et al., 2018), dialogue (Balakrishnan et al., 2019), summarization (Durmus et al., 2020), and question answering (Sellam et al., 2020). We summarize and describe the hallucination types for the QA task covered in this paper in Table 1. Benchmarks have been proposed to evaluate the extent to which LLMs hallucinate, such as Honovich et al. (2021), Huang et al. (2021), and Li et al. (2023).

**Hallucination detection and mitigation.** Methods that attempt to detect and mitigate hallucination without fact-checking include methods based on chain-of-thought (Wei et al., 2022; Huang et al., 2022; Dhuliawala et al., 2023), methods based on self-consistency (Wang et al., 2022; Huang et al., 2022; Mündler et al., 2023; Manakul et al., 2023), and methods that probe LLMs' hidden states (Azaria & Mitchell, 2023) or output probability distributions (Manakul et al., 2023). Since these methods do not augment LLMs with external knowledge, they usually struggle when LLMs' internal knowledge is inadequate. On the other hand, fact-checking-based methods (Roller et al., 2020; Komeili et al., 2021; Shuster et al., 2022a; 2021; 2022b; Izacard et al., 2022; Li et al., 2023; Semnani et al., 2023) rely on retrieved knowledge to prevent hallucinations in LLMs. However, these methods are often limited by the ways they retrieve knowledge and utilize the retrieved knowledge. For instance, Li et al. (2023) employ only a single query with the knowledge for detecting hallucinations in inputs that inherently necessitate multi-hop reasoning, which would benefit from a step-by-step query process. Semnani et al. (2023) instead introduces a robust framework initially designed to deliver fact-checked responses. This innovative method employs a comprehensive process that entails generating queries to fetch information from Wikipedia, summarizing and filtering the retrieved content, and then crafting a response informed by this vetted knowledge. However, when adapting WikiChat for hallucination detection, although it effectively uses retrieved knowledge for fact-checking in parallel, it occasionally neglects the coherence of the facts being verified, potentially leading to inaccuracies. Besides, during the fact-checking phase, WikiChat directly retrieves the evidence with the claim, which may not be effective when the claim itself is the result of hallucination as shown in Appendix B. Our approach,

in contrast, derives step-by-step queries with refined formulations and provides LLMs with either structured or unstructured knowledge for consecutive fact-checking, leading to better retrieval and higher knowledge utilization.

## 3 KNOWHALU

KnowHalu provides a systematic hallucination detection framework based on multi-step query and reasoning. It starts with "*Off-Target Hallucination Checking*" to pinpoint non-specific hallucinated answers, followed by "*Factual Checking*", verifying the correctness of the answer through a multi-step process based on different forms of knowledge.

### 3.1 OFF-TARGET HALLUCINATION CHECKING.

Current approaches in hallucination detection can mainly identify fabricated hallucinations, i.e., answers with mismatching facts (Manakul et al., 2023; Peng et al., 2023; Semnani et al., 2023). Yet, hallucinations also emerge in other types, as outlined in Table 1, which extend beyond simple factual inaccuracies. A typical trait of these off-target hallucinations is their factual correctness while a lack of direct relevance and helpfulness to the original query. For instance, given a question, "*What is the primary language in Barcelona?*", the hallucinated answer "*European languages*" is factually correct but fails in providing specific answers, thus being off-target, which is important for the quality of real-world LLMs. Note that we mainly focus on the potential hallucinations of LLMs in practice, so if the model rejects to answer a question, it will not be viewed as hallucinations. For example, if the answer to the question "What is the capital of France?" is "France is in Europe," it is categorized as an off-target hallucination due to a misinterpretation of the question. Conversely, responses like "Sorry, the question is hard" or "Sorry, I don't know" are not considered hallucinations in our work. These replies correctly interpret the question and reject to provide an answer, and thus do not fall under the "Misinterpretation of Question" category shown in Table 1. Instead, they acknowledge the difficulty in providing a specific answer, thereby offering a useful response. In fact, we should encourage models to refuse to answer questions that they do not know.

To bridge this gap, in KnowHalu, we first introduce a "*Off-Target Hallucination Checking*" phase, as depicted in the first row of Figure 1. This step aims to identify off-target hallucinations. A straightforward approach might involve prompting the language model to identify such hallucinations based on provided examples. However, this often results in high false positives, inaccurately flagging correct answers as hallucinations. To counter this, we address this challenge by solving an *extraction* task, which prompts the language model to extract specific entity or details requested by the original question from the answer. If the model fails to extract such specifics, it returns "NONE". This extraction-based specificity check is designed to reduce false positives while effectively identifying off-target hallucinations. Responses yielding "NONE" are directly labeled as hallucinations, and the remaining generations will be sent to the next phase for further factual checking. Examples of instructions in the extraction task for each type of off-target hallucination are provided in Appendix A.1.

### 3.2 FACTUAL CHECKING

The *Factual Checking* phase consists of five key steps: (a) *Step-wise Reasoning and Query* breaks down the original query into sub-queries following the logical reasoning process and generates different forms of sub-queries; (b) *Knowledge Retrieval* retrieves knowledge for each sub-query based on existing knowledge database; (c) *Knowledge Optimization* summarizes and refines the retrieved knowledge, and maps them to different forms, such as unstructured knowledge (object-replicate-object triplet); (d) *Judgment Based on Multi-form Knowledge* assesses the answer for each sub-query based on multi-form knowledge; and (e) *Aggregation* combines insights of judgments based on different forms of knowledge and makes a further refined judgment.

**a. Step-wise Reasoning and Query** In this step, we aim to break down the original query into local sub-queries following the reasoning logic, and we will retrieve knowledge for each sub-query sequentially (details in steps b and c), which is similar with ReAct (Yao et al., 2023), to cumulatively perform factual checking along the reasoning process. One main challenge here is "*how do we craft precise and effective sub-queries, which can accurately retrieve the relevant knowledge at each*

*logical step?*" To address this challenge, we identify two key factors that significantly enhance the accuracy of knowledge retrieval for factual checking: (1) continuous and direct (one-hop) queries, and (2) the formulation of queries. We will analyze these two key factors, which lead to our design choice below.

First, we observe that multi-hop queries (e.g., Example-1 in Figure 1) often struggle to retrieve specific and related knowledge due to their inherently complex and ambiguous context. On the other hand, the one-hop queries are effective to retrieve the most relevant and useful knowledge. Thus, we decompose the original query into sequence of simpler and direct one-hop sub-queries following the logical reasoning process, which significantly enhance the retrieval accuracy for factual checking. Concretely, this iterative querying process starts by interpreting the original query as a series of logical steps to form sub-queries accordingly, and then perform factual checking for each sub-query. For instance, based on the example in Figure 1, the initial query first confirms whether "*Star Wars*" is indeed the 1977 space-themed movie featuring Luke Skywalker. This step is crucial for connecting the movie to its composer. Subsequent queries delve deeper, examining the accuracy of other specific details provided in the answer, such as the composer's identity. The queries are intricately connected, each building upon the knowledge obtained from the previous one. This iterative process continuous to generate subsequent queries based on newly acquired knowledge, until the logical reasoning process is completed.

Second, we observe that the formulation of queries also plays a critical role for the final factual checking. In particular, queries with correct details will lead to high-quality knowledge retrieval; while queries with incorrect or unrelated entities may lead to poor and irrelevant knowledge retrieval. As a result, we propose two query formulations: *General Query* and *Specific Query*. The General Query avoids mentioning specific, potentially hallucinated details (e.g., "*Who composed the score for 'Star Wars'?*"); the Specific Query is constructed based on the key entities mentioned in the answers (e.g., "*Did Joy Williams compose the score for 'Star Wars'?*"), as shown in Figure 1. More concrete examples illustrating the impact of these two query formulations on the retrieval outcomes for both correct and hallucinated details can be found in Appendix B.

In our experiments shown in Section 5.2, we examine how different query formulations —*general* and *specific* — affect knowledge retrieval and the accuracy of final hallucination detection, by leveraging only one or both query formulations. More detailed prompts are provided in Appendix A.2.

**b. Knowledge Retrieval** We perform knowledge retrieval for each sub-query generated from step a. In particular, for QA tasks, we adopt the Retrieval-Augmented Generation (RAG) framework developed based on WikiPedia knowledge base (Semnani et al., 2023), and the retrieval is based on ColBERT v2 (Santhanam et al., 2022b) and PLAID (Santhanam et al., 2022a). We retrieve Top-K relevant passages for each sub-query, each formatted as "Title: ..., Article: ...". In addition, when we perform knowledge retrieval for summarization tasks, we treat the source document itself as the knowledge base for retrieval. In particular, we first segment the original documents into distinct text chunks. We then embed the sub-queries and text chunks into dense vectors using a text encoder. Similarly, we will retrieve the Top-K text chunks that exhibit the highest cosine similarity with the input sub-queries.

**c. Knowledge Optimization** The knowledge retrieved for each sub-query is usually a long and verbose passage with distracting irrelevant details. Thus, this step aims to leverage another LLM to distill useful information and optimize clear and concise knowledge, which could be in different forms. In particular, we propose two forms of knowledge, *unstructured* and *structured* knowledge. The unstructured knowledge represents the texts retrieved from given knowledge bases in a concise way, such as "*'Star Wars,' released in 1977, is the space-themed movie in which the character Luke Skywalker first appeared.*" Since the unstructured text may not be precise for logical reasoning, we also retrieve structured knowledge as object-predicate-object triplets, such as *("Star Wars", was, 1977 space-themed movie)* and *(Luke Skywalker, first appeared in, "Star Wars")* (examples of our demonstrations are in Appendix A.3). Such multi-form knowledge will effectively assist LLMs to perform logical reasoning and final factual checking. In addition, if a query retrieves no relevant knowledge, the LLM is instructed to respond with "*No specific information is available*".

**d. Judgment Based on Multi-form Knowledge** After obtaining the retrieved multi-form knowledge for sub-queries, we gather *#Query#* and *#Knowledge#* and present them to another LLM for hallucination judgment. The *#Judgment#* assesses the sub-query and its corresponding knowledge sequentially to ascertain if there is any contradiction to verification each detail in the answer. If there

Table 2: Performance of different methods for hallucination detection in QA task. Results of methods using external ground truth knowledge (i.e., knowledge provided by HaluEval) are shown inside the parentheses, and results generated based on Wiki knowledge are shown outside the parentheses.

| Model | Method | TPR (%) | TNR (%) | Avg Inconclusive Rate (%) | Avg Acc (%) |
|---|---|---|---|---|---|
| GPT-4 | Zero-Shot CoT | 68.3 | 61.8 | – | 65.05 |
| Starling-7B | SelfCheckGPT (Manakul et al., 2023) | 89.7 | 30.3 | – | 60.00 |
| | HaluEval (Vanilla) (Li et al., 2023) | 33.2 | 80.3 | – | 56.75 |
| | HaluEval (CoT) (Li et al., 2023) | 68.7 | 26.0 | – | 47.35 |
| | HaluEval (Knowledge) (Li et al., 2023) | 33.0 (82.0) | 60.3 (40.0) | – | 46.65 (61.00) |
| | Self Consistency (Wang et al., 2022) | 37.5 (80.1) | 58.6 (41.8) | – | 48.05 (60.95) |
| | KnowHalu (Structured) | 68.1 (67.8) | 65.6 (83.1) | 10.50 (8.45) | 66.85 (75.45) |
| | KnowHalu (Unstructured) | 68.2 (72.4) | 69.9 (85.9) | **6.75 (4.85)** | 69.05 (79.15) |
| | KnowHalu (Aggregation) | **68.7 (72.7)** | **75.9 (88.7)** | **6.75 (4.85)** | **72.30 (80.70)** |
| Mistral-7B | SelfCheckGPT (Manakul et al., 2023) | 67.9 | 45.9 | – | 56.90 |
| | HaluEval (Vanilla) (Li et al., 2023) | 40.9 | 41.3 | – | 41.10 |
| | HaluEval (CoT) (Li et al., 2023) | 48.0 | 41.1 | – | 44.55 |
| | HaluEval (Knowledge) (Li et al., 2023) | 50.9 (55.1) | 47.9 (11.1) | – | 49.40 (33.10) |
| | Self Consistency (Wang et al., 2022) | 51.9 (55.0) | 47.6 (10.9) | – | 49.75 (32.95) |
| | KnowHalu (Structured) | 54.0 (59.0) | 67.3 (82.6) | 12.35 (9.45) | 60.65 (70.80) |
| | KnowHalu (Unstructured) | 63.5 (62.5) | 58.4 (82.2) | 11.70 (6.90) | 60.95 (72.35) |
| | KnowHalu (Aggregation) | **60.7 (63.3)** | **65.4 (85.1)** | **10.80 (6.20)** | **63.05 (74.20)** |
| GPT-3.5 | WikiChat (Semnani et al., 2023) | 16.0 | 82.2 | – | 49.10 |
| | SelfCheckGPT (Manakul et al., 2023) | 67.8 | 61.8 | – | 64.80 |
| | HaluEval (Vanilla) (Li et al., 2023) | 44.1 | 55.2 | – | 48.15 |
| | HaluEval (CoT) (Li et al., 2023) | 66.5 | 21.6 | – | 44.05 |
| | HaluEval (Knowledge) (Li et al., 2023) | 34.4 (38.1) | 71.7 (75.7) | – | 53.05 (56.90) |
| | Self Consistency (Wang et al., 2022) | 37.1 (33.6) | 64.7 (76.3) | – | 50.90 (54.95) |
| | KnowHalu (Structured) | 72.6 (75.7) | 66.6 (80.0) | 6.60 (7.10) | 69.60 (77.85) |
| | KnowHalu (Unstructured) | 77.3 (68.9) | 53.2 (75.7) | 11.90 (14.50) | 65.25 (72.30) |
| | KnowHalu (Aggregation) | **76.3 (77.5)** | **67.8 (83.1)** | **6.60 (7.05)** | **72.05 (80.30)** |

is any conflict between the answer the knowledge from a sub-query, the judgment is **INCORRECT**. On the other hand, if all the details of the answer are verified by the knowledge of sub-queries, the judgment is **CORRECT**. For some scenarios, where the knowledge is inadequate for a conclusive judgment, the output will be **INCONCLUSIVE**. Prompts used to guide this judgment process are shown in Appendix A.4.

**e. Aggregation** The judgment of hallucination above is based on each form of the retrieved knowledge (e.g., structured and unstructured). To mitigate the prediction uncertainty, here we aggregate these judgment based on the multi-form knowledge to make the final prediction.

The motivation for this aggregation mechanism is two-fold: 1) each knowledge form might uniquely identify cases that the other cannot, particularly when one yields an INCONCLUSIVE judgment and the other does not; 2) when the LLM makes a hallucinated judgment, it may lead to low confidence scores for the judgment of "CORRECT" or "INCORRECT." Thus, we can rely on the alternative knowledge form if it demonstrates a higher confidence for the judgment.

Concretely, we treat the judgment based on one form of the knowledge—typically the form yielding better average accuracy—as the *base judgment* and that of the other forms of knowledge as *supplement judgment*. If the confidence score for the base judgment falls below the a specific threshold $\delta_1$ and the supplement judgment maintains higher confidence above $\delta_2$, we will take the supplement judgment as the final prediction. In all other cases, the base judgment will perform as the final prediction. The corresponding pseudo-code and more details for this aggregation mechanism are provided in Appendix E.3.

## 4 EXPERIMENTS

We have evaluated KnowHalu on the standard HaluEval dataset (Li et al., 2023), comparing with SOTA hallucination detection baselines under different settings. We find that 1) KnowHalu consistently outperforms the baselines in terms of hallucination detection in different tasks, 2) different models benefit differently from knowledge forms; GPT-3.5 performs better with structured knowledge, while open-sourced models like Starling-7B or Mistral-7B are more effective with unstructured knowledge; 3) the aggregation of predictions from different knowledge forms can further improve detection accuracy. All experiments were conducted on a single NVIDIA A6000 GPU.

### 4.1 EXPERIMENTAL SETUP

**Dataset.** We conduct our experiments using the standard HaluEval dataset (Li et al., 2023), and focus on hallucination detection for two primary tasks: multi-hop QA and text summarization. For the

Table 3: Performance of different methods for hallucination detection in Text Summarization.

| Model | Method | TPR (%) | TNR (%) | Avg Acc (%) |
|---|---|---|---|---|
| GPT-4 | Zero-Shot CoT | 43.0 | 83.0 | 63.0 |
| Starling-7B | SelfCheckGPT (Manakul et al., 2023) | 80.2 | 38.4 | 59.3 |
| | HaluEval (Vanilla) (Li et al., 2023) | 17.4 | 95.6 | 56.5 |
| | HaluEval (CoT) (Li et al., 2023) | 31.6 | 81.0 | 56.3 |
| | Self Consistency (Wang et al., 2022) | 24.0 | 89.0 | 56.5 |
| | KnowHalu (Structured) | 80.2 | 45.4 | 62.8 |
| | KnowHalu (Unstructured) | 65.0 | 67.2 | 66.1 |
| | KnowHalu (Aggregation) | 59.2 | 75.5 | 67.3 |
| Mistral-7B | SelfCheckGPT (Manakul et al., 2023) | 29.0 | 91.8 | 60.4 |
| | HaluEval (Vanilla) (Li et al., 2023) | 79.0 | 9.0 | 44.0 |
| | HaluEval (CoT) (Li et al., 2023) | 84.8 | 6.0 | 45.4 |
| | Self Consistency (Wang et al., 2022) | 83.4 | 6.6 | 45.0 |
| | KnowHalu (Structured) | 68.6 | 63.2 | 65.9 |
| | KnowHalu (Unstructured) | 67.0 | 67.4 | 67.2 |
| | KnowHalu (Aggregation) | 67.0 | 67.8 | 67.4 |
| GPT-3.5 | SelfCheckGPT (Manakul et al., 2023) | 33.6 | 86.8 | 60.2 |
| | HaluEval (Vanilla) (Li et al., 2023) | 66.6 | 58.0 | 62.3 |
| | HaluEval (CoT) (Li et al., 2023) | 44.4 | 63.4 | 53.9 |
| | Self Consistency (Wang et al., 2022) | 66.6 | 59.0 | 62.8 |
| | KnowHalu (Structured) | 64.4 | 71.0 | 67.7 |
| | KnowHalu (Unstructured) | 62.8 | 68.0 | 65.4 |
| | KnowHalu (Aggregation) | 69.0 | 68.0 | 68.5 |

Table 4: Performance of KnowHalu using different query formulations, evaluated based on the Starling-7B model.

| Query | Method | TPR | ARP | TNR | ARN | Avg Acc (%) |
|---|---|---|---|---|---|---|
| Specific Query | KnowHalu (Structured) | 57.4 | 19.8 | 64.1 | 18.6 | 60.75 |
| | KnowHalu (Unstructured) | 66.0 | 11.5 | 64.9 | 10.7 | 65.45 |
| General Query | KnowHalu (Structured) | 65.6 | 15.5 | 58.7 | 18.9 | 62.15 |
| | KnowHalu (Unstructured) | 70.4 | 10.6 | 60.7 | 15.9 | 65.55 |
| Combined Queries | KnowHalu (Structured) | 68.1 | 9.0 | 65.6 | 12.0 | 66.85 |
| | KnowHalu (Unstructured) | 68.2 | 5.3 | 69.9 | 8.2 | 69.05 |

Table 5: Performance of KnowHalu using different number of retrieved Wiki passages $K$, evaluated based on the Starling-7B model.

| Top-$K$ Passages | Method | TPR | ARP | TNR | ARN | Avg Acc (%) |
|---|---|---|---|---|---|---|
| $K = 1$ | KnowHalu (Structured) | 61.1 | 16.1 | 64.3 | 16.6 | 62.70 |
| | KnowHalu (Unstructured) | 65.6 | 10.4 | 64.8 | 13.0 | 65.20 |
| $K = 2$ | KnowHalu (Structured) | 68.1 | 9.0 | 65.6 | 12.0 | 66.85 |
| | KnowHalu (Unstructured) | 68.2 | 5.3 | 69.9 | 8.2 | 69.05 |
| $K = 3$ | KnowHalu (Structured) | 67.2 | 7.8 | 66.2 | 9.9 | 66.70 |
| | KnowHalu (Unstructured) | 68.6 | 4.1 | 70.8 | 4.5 | 69.70 |
| $K = 4$ | KnowHalu (Structured) | 67.1 | 8.0 | 65.8 | 11.1 | 66.45 |
| | KnowHalu (Unstructured) | 68.8 | 3.7 | 69.6 | 5.2 | 69.20 |
| $K = 5$ | KnowHalu (Structured) | 64.2 | 9.9 | 67.3 | 9.3 | 65.75 |
| | KnowHalu (Unstructured) | 66.9 | 4.3 | 72.7 | 4.0 | 69.80 |

multi-hop QA task, the dataset comprises questions and correct answers from HotpotQA (Yang et al., 2018), with hallucinated answers generated by ChatGPT. In the text summarization task, the dataset includes documents and their non-hallucinated summaries from CNN/Daily Mail (See et al., 2017), along with hallucinated summaries generated by ChatGPT.

In our experiment, we randomly sample $1,000$ pairs from the QA task as the test set. Each test pair comprises both a correct answer and a hallucinated answer to the same question. Additionally, we sampled 500 pairs from the summary task, with each pair containing both accurate and hallucinated counterparts for the same document. We use these balanced test sets for evaluation and comparison.

**Baselines.** For the QA hallucination detection task, our study evaluates seven state-of-the-art baselines, each chosen to demonstrate distinct aspects of model performance in detecting hallucinations. (1) The first three baselines from the HaluEval suite (Li et al., 2023)—*HaluEval (Vanilla)*, which makes judgments without external knowledge; *HaluEval (Knowledge)*, which utilizes external knowledge; and *HaluEval (CoT)*, employing Chain-of-Thought reasoning—are specifically tailored for the dataset we used, ensuring optimized performance on the HaluEval benchmark. (2) *GPT-4 (Zero-shot CoT)* leverages the intrinsic world knowledge of a model dated 2023-11-06, which is the same date as the wiki database we use for retrieval, testing whether a smaller model with systematic knowledge reasoning can outperform a larger model based on its extensive pre-trained knowledge. (3) *WikiChat* (Semnani et al., 2023) generates responses by retrieving and summarizing Wikipedia information, using the same data as our wiki database for retrieval, to ensure accuracy through fact-checking. (4) *SelfCheckGPT* (Manakul et al., 2023) uses the inherent knowledge of large language models to generate and evaluate multiple responses for consistency without external data. (5) *Self Consistency* (Wang et al., 2022) involves sampling hallucination judgments from the three HaluEval baselines multiple times and taking a majority vote; we apply it on all three HaluEval baselines and report the best outcome. For *SelfCheckGPT* and *Self Consistency*, we sample 20 times using a temperature of 1.0. The prompts employed to query *GPT-4 (Zero-shot CoT)*, *WikiChat*, and *SelfCheckGPT* for hallucination detection are in Appendix A.5.

**Models.** For our experiments, we use three models: Starling-7B (`Starling-LM-7B-alpha`) (Zhu et al., 2023) and Mistral-7B (`Mistral-7B-Instruct-v0.2`) (Jiang et al., 2023), which are two open-source models that have shown high performance on the LMSYS Chatbot Arena Leaderboard (LMSYS, 2023; Zheng et al., 2023); and GPT-3.5 (`gpt-3.5-turbo-1106`) (OpenAI, 2023), a closed-source model from OpenAI.

**Metric.** Our evaluation focuses on five key metrics: True Positive Rate (TPR), True Negative Rate (TNR), Average Accuracy (Avg Acc), and Average Inconclusive Rate. TPR quantifies the ratio of correctly identified hallucinations, TNR measures the ratio of correctly identified non-hallucinations, and Avg Acc denotes the overall accuracy. Average Inconclusive Rate represents the model capability of identifying *inconclusive* cases. Note that it is not always possible to successfully retrieve the corresponding knowledge to verify the answer. However, existing baselines based on external knowledge still require the model to provide a binary judgment (Yes/No); thus, the accuracy reported could be higher than their actual performance since some answers actually cannot be assessed with

the available knowledge. On the contrary, KnowHalu allows the INCONCLUSIVE option to provide more informative judgments based on our capable framework. In this way, the average accuracy metric is slightly unfair for KnowHalu since we aim to provide a more fine-grained detection; otherwise, its reported average accuracy should be even higher. Nevertheless, KnowHalu still beats all baselines significantly in terms of the average accuracy (Table 2).

## 4.2 HALLUCINATION DETECTION ON QA TASK

**Setup.** To detect hallucinations in the QA Task, we test two distinct knowledge sources. The first, which we refer to as "*off-the-shelf knowledge*," is the knowledge provided in the HaluEval dataset. This consists of specific passages that are directly related to the question-answer pairs within the dataset, which serves as a natural upper bound for the quality of retrieved knowledge. The second knowledge source, which we refer to as "*Wiki retrieval knowledge*," comes from the information retrieval system constructed over the Wikipedia Database as outlined in step (b) in Section 3.2. This system fetches the Top-K most relevant passages in response to a given query. We aim to evaluate the effectiveness of different hallucination detection approaches given these two knowledge sources.

When utilizing the "*Wiki retrieval knowledge*", the number of fetched passages $K$ is consistently set to 2 for all methods. For our method, we report the results considering different query formulations. A detailed analysis of the influence of different query formulations and $K$ is presented in Section 5.2 and Section 5.3, respectively.

**Results.** The main results are shown in Table 2. We observe that leveraging our sequential reasoning and query approach, coupled with a well-formulated query for knowledge retrieval and the aggregation of two distinct forms of knowledge, KnowHalu consistently outperforms baselines by around $15\%$ when using the same knowledge source with the same model.

Furthermore, our results reveal several intriguing observations: (1) relying solely on the pre-trained knowledge of LLMs and employing multiple samples to detect contradictions between different samples (as in SelfCheckGPT and Self Consistency) typically results in sub-optimal performance; (2) employing systematic, step-wise reasoning and querying enables a small 7B model (Starling-7B) within KnowHalu to achieve comparative performance with GPT-3.5; (3) using all the three samller models in KnowHalu demonstrates superior detection performance when compared to the powerful GPT-4, which has implicit reasoning capabilities and knowledge; (4) the form of knowledge matters for different models—open-sourced models like Starling-7B and Mistral-7B appear to perform better with unstructured knowledge, while GPT-3.5 seems to benefit more from structured knowledge (i.e., triplets), enhancing the need for aggregation mechanisms. A comprehensive analysis of the individual contributions of each component within KnowHalu is in Section 5.

## 4.3 HALLUCINATION DETECTION ON SUMMARIZATION TASK

**Setup.** In the task of text summarization, the original document serves as the primary source of knowledge. During our experiments, we segment the document into passages with fewer than $40$ words each. Both the input query and retrieved passages are encoded using BGE large model (BAAI, 2023) from FlagEmbedding (Xiao et al., 2023b; Zhang et al., 2023a; Xiao et al., 2023a). For each query, the Top-K relevant passages are retrieved for knowledge optimization, with the number of passages $K$ set to 3. The impact of varying $K$ is further analyzed in Section 5.3.

Unlike the QA task, to detect hallucinations in the summarization task, any detail in the summary that cannot be supported or identified in the original document will be considered as a hallucination, which means we have a complete knowledge source. As a result, the off-target hallucination checking phase is not required for this task, allowing us to move directly to factual checking. The judgment now only includes CORRECT and INCORRECT, as cases that would be classified as INCONCLUSIVE in the QA task are inherently INCORRECT in text summarization task. In addition, given that some summaries are quite lengthy, we segment the original summary into small parts, with each segment comprising no more than 30 words. Each segment is independently evaluated for hallucination, and the entire summary is labeled as a hallucination if any part receives an INCORRECT judgment.

**Results.** The main results are presented in Table 3. We observe that KnowHalu significantly outperforms the baselines, achieving performance increases of $8.0\%$ with Starling-7B, $7.0\%$ with Mistral-7B, and $6.2\%$ with GPT-3.5. Notably, we can see that nearly all variations of KnowHalu surpass the powerful GPT-4, demonstrating a superior performance. In particular, GPT-3.5 model demonstrates a great advantage when utilizing structured knowledge, whereas both the Starling-7B and Mistral-7B models still benefit more from unstructured knowledge. Besides, as we can see, reliance on detecting contradictions between different sample summarizations (SelfCheckGPT) typ-

Table 6: Impact of Off-Target Hallucination Checking on Hallucination Detection Performance using the Starling-7B Model for QA Task. The table reports the True Positive Rate (TPR), True Negative Rate (TNR), and Average Accuracy (Avg Acc) for each method.

| Method | TPR (%) | TNR (%) | Avg Acc (%) |
|---|---|---|---|
| HaluEval (Knowledge) | 33.0 (82.0) | 60.3 (40.0) | 46.65 (61.00) |
| Pure Fact-Checking (Structured) | 52.1 (48.2) | 67.2 (86.6) | 59.65 (67.40) |
| **+ Off-Target Hallucination Checking** | **68.1 (67.8)** | **65.6 (83.1)** | **66.85 (75.45)** |
| Pure Fact-Checking (Unstructured) | 53.4 (56.0) | 71.7 (88.1) | 62.55 (72.05) |
| **+ Off-Target Hallucination Checking** | **68.2 (72.4)** | **69.9 (85.9)** | **69.05 (79.15)** |

ically results in highly unbalanced TPR and TNR. For example, with GPT-3.5, it yields a TNR of $86.8\%$ but a corresponding TPR of only $33.6\%$. At the same time, relying solely on the majority sample judgment of hallucinations (Self-Consistency) offers only marginal improvements. Additionally, we also observe some 'lazy' behaviors in GPT-3.5 during step-wise reasoning, as demonstrated in Appendix D. We also present the performance of various temporal versions of GPT-3.5 in Appendix C for comprehensive explorations.

## 5 ABLATION STUDIES

### 5.1 IMPACT OF OFF-TARGET HALLUCINATION CHECKING

Prompts provided by HaluEval (Li et al., 2023) not only cover cases of fabrication hallucination but also off-target hallucinations as shown in Table 1. Thus, our pipeline separates the process into two phases: (1) treating the detection of off-target hallucinations as an independent task; (2) if an off-target hallucination is detected, no further checking is required; otherwise, we proceed to a second-phase for factual checking. This approach raises two intriguing questions: (1) whether such decomposition improves hallucination detection performance, and (2) what is the performance when factual checking is conducted directly without any preliminary off-target hallucination demonstration. As shown in Table 6, incorporating off-target checking consistently enhances the detection of hallucinated cases, with an approximately $15\%$ improvement in TPR and $2\%$ in FPR. Furthermore, even without off-target checking, our standalone factual checking phase still surpasses the baseline using the same knowledge source, demonstrating the effectiveness of our multi-query and reasoning process.

### 5.2 FORMULATIONS OF QUERIES

We investigate the impact of different query formulations used for knowledge retrieval on hallucination detection. All experiments are conducted using Starling-7B with $K = 2$ for Wiki retrieval knowledge. We evaluate the following three approaches: (1) using only *specific queries*, (2) using only *general queries*, and (3) combining the Top-K results from both query types. The results are detailed in Table 4. As we can see, the formulation of the query is crucial in knowledge retrieval. In particular, *specific query* formulation enhances the accuracy for non-hallucinated cases but reduces that for hallucinated ones due to polluted context. Conversely, using *general queries* yields an inverse effect. Combining both query types improves the overall detection accuracy by at least $3.5\%$ and reduces the abstention rate by over $5\%$, demonstrating that the combination of both query formulations indeed leads to more accurate and relevant knowledge retrieval. The results conducted with the off-the-shelf knowledge and the results for text summarization are presented in Appendix E.1.

### 5.3 NUMBER OF RETRIEVAL KNOWLEDGE

We explore how the number of retrieved Wiki passages, $K$, impacts the performance of hallucination detection in this section. Throughout this analysis, we consistently employ a combination of both specific and general queries for knowledge retrieval, focusing on assessing the influence of varying $K$. The results are presented in Table 5. We can observe that increasing the number of retrieved passages enhances detection accuracy and reduces the abstain rate. In addition, the performance converges when $K$ is greater than 2, and additional knowledge will only provide marginal improvement, highlighting the potential efficiency of KnowHalu. We also provide the results for similar experiments conducted on text summarization in Appendix E.2.

### 5.4 AGGREGATION BASED ON MULTI-FORM KNOWLEDGE

We further explore mitigating hallucinations by implementing a confidence-based aggregation mechanism that utilizes various forms of knowledge, both structured and unstructured. The motivation of our approach is the observation that judgments susceptible to hallucinations typically have lower confidence levels compared to those that are accurate and free from hallucinations. Consequently,

Table 7: Latency per instance (in seconds) for various methods on the QA task using the Starling-7B model on one A6000 GPU.

| Method | Knowledge Source | Latency (s/instance) |
|---|---|---|
| HaluEval (Knowledge) | Off-the-shelf | 0.27 |
| HaluEval (Knowledge) | Wiki | 0.57 |
| Self Consistency | Off-the-shelf | 4.36 |
| Self Consistency | Wiki | 8.71 |
| KnowHalu | Off-the-shelf | 7.98 |
| KnowHalu | Wiki | 8.75 |
| WikiChat | Wiki | 10.62 |
| SelfCheckGPT | - | 23.89 |

we adopt a strategy where if a base judgment, derived from one form of knowledge, displays low confidence (below $\delta_1$), and a supplementary judgment from a different form of knowledge shows significantly higher confidence (above $\delta_2$), the latter is prioritized based on its reliability.

To select thresholds $\delta_1$ and $\delta_2$, we employ a data-driven approach that utilizes the quantile of the confidence distribution associated with each form of knowledge. We achieve this by using a small validation set for both tasks, during which we collect confidence distributions for judgments obtained based on each knowledge type. This approach facilitates a more precise evaluation of $\delta_1$ and $\delta_2$ through an examination of various quantiles within these distributions. By adjusting $\delta_1$ and $\delta_2$ according to these quantiles on the validation set, we aim to identify the optimal configurations that yield the highest average accuracy for each task. In our experiments, we consistently utilize the judgments based on the form of knowledge that provides the best average accuracy as the base judgment. The specific values of $\delta_1$ and $\delta_2$, along with a detailed description of the process for selecting them, are provided in Appendix E.3.

### 5.5 LATENCY COMPARISON

Our reasoning process, while sequential across multiple steps, benefits from shared key-value caching. We retain the cache from previous reasoning steps to accelerate subsequent ones, thereby reducing latency. Specifically, the latency for the methods using the Starling-7B model on one A6000 GPU per instance for the QA task is shown in Table 7. As we can see, while our method shows a slight increase in latency compared to the sample-based method Self Consistency (Wang et al., 2022) when using off-the-shelf knowledge, it incurs similar costs when using wiki-retrieved knowledge. Notably, our method consistently outperforms Self Consistency by at least 20%, regardless of the underlying LLM used, as shown in Table 2. Additionally, our method's latency is lower than that of WikiChat (Semnani et al., 2023)—which extracts and verifies claims individually—and is significantly lower than that of another sample-based hallucination detection method, SelfCheckGPT (Manakul et al., 2023). Consequently, the latency introduced by our approach remains within acceptable limits, staying under 10 seconds per instance for detecting hallucinations in answers to multi-hop complex questions, such as those in the HaluEval dataset (based on HotpotQA) used in our experiments. Moreover, latency is expected to be significantly lower for simpler, one-hop questions, making our method a viable option in settings where higher accuracy is essential.

### 6 CONCLUSION

In this work, we have introduced KnowHalu, a novel framework for detecting hallucinations in text generated by Large Language Models (LLMs). Our approach stands out by employing a two-phase process: off-target Hallucination Checking and Factual Checking, which includes multi-form knowledge retrieval and optimization, along with an aggregation method for the final judgment. Through extensive experimentation on standard datasets, KnowHalu has demonstrated significant improvements over state-of-the-art baselines in detecting hallucinations in both QA and text summarization tasks. While the current implementation has limitations in handling extended dialogues and much longer responses, specifically the lack of correlation consideration between sentences within a single response, future extensions will focus on adapting the framework for dialogue systems and optimizing it for longer interactions.

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

# A PROMPTS

## A.1 PROMPT FOR OFF-TARGET HALLUCINATION CHECKING

For each type of hallucination listed in Table 1, we include one to two illustrative examples in the prompts for instructing the LLMs to implement the extraction task as shown in Table 8. This results in a compilation of 10 examples where the extraction outcome is "NONE," indicative of off-target hallucinations. Additionally, we include 11 examples where the extraction successfully identifies the entity pertinent to the question. It's important to note that these 11 examples feature a mix of correct and incorrect (fabricated) answers. This variety is intentional, as the primary objective in this phase is to filter out off-target hallucinations, thus allowing for the possibility of incorrect (fabricated) responses in the examples.

## A.2 PROMPT FOR STEP-WISE REASONING AND QUERY

We present examples of prompts for Step-wise Reasoning and Query, illustrating our approach with both structured and unstructured knowledge forms for the QA task. These examples are detailed in Table 9 for structured knowledge and in Table 10 for unstructured knowledge, respectively. Specifically, we provide three examples of correct answers, three examples of hallucinated answers, and two examples where the knowledge is not available during the reasoning and query process. Besides, note that the prompt examples presented here are designed for combined queries. Therefore, in the demonstration of the line for *#Query#*, it starts with a specific query followed by a general query enclosed in brackets. This indicates that the LLMs are instructed to produce queries in this composite format, effectively integrating both specific and general inquiry approaches within a single output. During the knowledge retrieval step, both types of queries are employed for retrieval, and the retrieved passages for each are merged. In scenarios where only a specific query is adopted, the prompt will be adjusted to retain only the initial specific query, omitting the bracketed portion. Conversely, for general queries, the specific query is excluded, maintaining only the general query. The prompt structure for the text summarization task follows a similar format, also featuring three examples each for correct and hallucinated answers. Therefore, we only display the QA task prompts as representative examples to avoid redundancy.

## A.3 PROMPT FOR KNOWLEDGE OPTIMIZATION

In our experiments, we extract related knowledge from either *Wiki retrieval knowledge* or *off-the-shelf knowledge* in an optimized format, corresponding to either structured knowledge (in triplets) or unstructured knowledge (in normal semantic text). For the QA task, examples of prompts based on Wiki retrieval knowledge, tailored for these two forms, are illustrated in Table 11 and Table 12, respectively. Given the similarity in format for off-the-shelf knowledge on QA task and the approach used for the Text Summarization task, we omit those details to prevent redundancy.

## A.4 PROMPT FOR JUDGMENT

Upon completing the step-wise reasoning and query process, we accumulate sufficient information to proceed to the final judgment for each form of knowledge. At this stage, the *#Thought#* line, initially instrumental in guiding the query, becomes redundant and is therefore removed. We retain only the *#Query#* and *#Knowledge#* inputs for the LLM to facilitate the final judgment. The prompts used for this purpose on QA task, tailored to both structured and unstructured knowledge, are displayed in Table 13 for structured knowledge and in Table 14 for unstructured knowledge. Given the prompts for the text summarization task follow a similar format, we also omit their detailed presentation here to avoid repetition.

## A.5 SETUP DETAILS FOR BASELINES

### A.5.1 GPT-4 WITH ZERO-SHOT COT

**QA.** For GPT-4 with zero-shot reasoning, we utilize the system prompt from HaluEval:

```
You are a hallucination detector. You MUST determine if the provided
answer contains hallucination or not for the question based on the world
knowledge. The answer you provided MUST be "Yes" or "No". You should
first provide your judgment and then provide your reasoning steps.
```

And the corresponding user prompt is:

```
You should try your best to determine if the answer contains non-factual
or hallucinated information. The judgement you give MUST be "Yes" or "No".
You should first provide your judgment and then provide your reasoning
steps.

#Question#: {question}
#Answer#: {answer}
#Your Judgement#:
```

**Text Summarization.** We utilize the system prompt from HaluEval:

```
You are a summary judge. You MUST determine if the provided summary
contains non-factual or hallucinated information. The answer you give
MUST be "Yes" or "No". You should first provide your judgment and then
provide your reasoning steps.
```

And the corresponding user prompt is:

```
You should try your best to determine if the summary contains non-factual
or hallucinated information for the document. The judgement you give MUST
be "Yes" or "No". You should first provide your judgment and then provide
your reasoning steps.

#Document#: {document}
#Summary#: {summary}
#Your Judgement#:
```

### A.5.2 SELFCHECKGPT

**QA.** We start by prompting the underlying large language model (LLM) to generate an answer to the given question, and similarly with the SelfCheckGPT (Manakul et al., 2023), we sampled the answer 20 times at a temperature setting of 1.0. Next, we employ a prompt adapted from SelfCheckGPT (Manakul et al., 2023), which compares these 20 newly generated answers against the original reference answer provided in HaluEval:

```
Question: {question}
Answer-1: {original_answer}
Answer-2: {newly_sampled_answer}

Is the Answer-1 supported by the Answer-2 for the same question mentioned
above? Provide the judgment with Yes or No.

Judgment:
```

The final judgment of hallucination is based on the majority vote from these comparisons.

**Text Summarization.** Similarly, we begin by prompting the underlying large language model (LLM) to generate a summary of the original document, and sample this summary 20 times at a temperature setting of 1.0. We then utilize a prompt adapted from SelfCheckGPT (Manakul et al., 2023), which assesses the hallucinations in the original summary provided in HaluEval against these 20 newly generated summaries. Given that summaries are typically lengthy, we follow the same protocol used in both KnowHalu and SelfCheckGPT by segmenting the original summary into smaller parts, each comprising no more than 30 words. Each segment is evaluated against the newly sampled summaries (used as the context) to determine the presence of hallucinations:

```
Context: {context}
Sentence: {segment}
```

```
Is the sentence supported by the context above? Answer Yes or No.

Judgment:
```

The hallucination judgment for each segment is based on the majority vote among the 20 new generated summaries. If any segment of the original summary is identified as a hallucination, the entire summary is labeled as such, consistent with the setting for KnowHalu.

### A.5.3 WIKICHAT

For WikiChat (Semnani et al., 2023), originally crafted for generating factual responses rather than conducting hallucination checks, we adapt its fact-checking capability. We prompt the LLM to articulate claims regarding the hallucination status of an answer, followed by detailed reasoning. This approach enables WikiChat's fact-checking mechanism to verify these claims, facilitating the generation of accurate judgments.

The user prompt employed to elicit reasoned judgments and claims is outlined as follows:
```
#Question#: {question} #Answer#: {answer} Is this answer to the question
a result of hallucination? Please indicate your judgment as either "Yes"
or "No," accompanied by a step-by-step explanation."
```
In leveraging WikiChat for hallucination detection, we utilize its comprehensive workflow, initially designed for factual checking. This includes generating a query to fetch information from Wikipedia, summarizing and filtering retrieved passages, and utilizing an LLM for initial response generation. By prompting the LLM to articulate judgment alongside claim statements, we engage WikiChat's fact-checking mechanism against the retrieved evidence, thereby refining the response through iterations. This process guarantees that the final determination of hallucination status is both evidence-based and meticulously refined for precision. Additionally, we illustrate the detailed process by which WikiChat generates the final fact-checked judgment in Table 15, providing insight into the iterative refinement and fact-checking stages integral of WikiChat.

## B  ILLUSTRATIVE CASES OF RETRIEVAL OUTCOMES BASED ON QUERY FORMULATION VARIATIONS

In this section, we present a range of illustrative cases that showcase the outcomes of retrieval based on the specific nuances in query formulation. This includes both specific and general queries aimed at verifying correct and hallucinated details. These examples utilize the same retrieval system built upon the Wikipedia database, as constructed in WikiChat (Semnani et al., 2023). The objective is to subjectively demonstrate the differences in retrieval effectiveness between these two types of queries. We display the top two results of each retrieval instance. The complete set of example cases is detailed in Table 16.

The findings indicate that specific queries, by leveraging additional information from the knowledge base, can yield highly accurate results when verifying correct details. However, their effectiveness diminishes when applied to hallucinated details, often leading to the retrieval of irrelevant or nonexistent information. In contrast, general queries excel in retrieving better results for hallucinated details, as they avoid the pitfalls of specific, possibly inaccurate information. Yet, this approach does not capitalize on the extra information provided in the answer, which can result in less effective verification of correct details. Therefore, each query formulation has its distinct advantages and limitations, necessitating a strategic choice based on the nature of the detail being verified.

## C  PERFORMANCE COMPARISON FOR DIFFERENT GPT-3.5 VERSIONS IN QA TASK

To examine the impact of different GPT-3.5 model versions, we tested `gpt-3.5-turbo-0613` and `gpt-3.5-turbo-1106` on the QA Task, with results presented in Table 17. Consistent with observations made by Chen et al. (2023) in HotpotQA (Yang et al., 2018), newer versions of GPT-3.5 exhibit a decline in performance. For instance, when simply prompting GPT-3.5 to answer questions without Chain of Thought (CoT) or knowledge support, akin to HaluEval (Vanilla), the performance drops by approximately 12%. In addition, with the integration of CoT (HaluEval(CoT)), the performance with the newer version drops 6.90%, and with knowledge-enhanced prompts (HaluEval(Knowledge)), the decrease is 10.75%.

However, interestingly, within our framework KnowHalu, the performance instead slightly improves by 1% to 2%, suggesting that our approach can leverage the inherent capabilities of the GPT-3.5 model more efficiently. Moreover, we observe that GPT-3.5 models across both versions particularly benefit more from structured knowledge forms, i.e., using triplets for reasoning, indicating their proficiency in managing structured knowledge within the QA task explored here, which is also one of the main findings in our KnowHalu framework.

## D    FAILURE CASES FOR GENERATING QUERY WITH GPT-3.5 IN SUMMARIZATION TASK

We find the performance of GPT-3.5 used in KnowHalu could have high uncertainty. This is primarily due to two key observations during our experiments: first, GPT-3.5 occasionally rejects certain summaries, which are recognized with inappropriate or potentially invasive content, an issue not analyzed in baselines; second, GPT-3.5 tends to demonstrate 'lazy' behavior during the step-wise querying process. For instance, it may prematurely conclude that a summary does not necessitate further verification, or it might simply conclude the correctness of a summary even without generating any query.

In specific, we first encountered instances where the model exhibited reluctance in generating queries, often citing concerns related to information privacy or involving sensitivity:

```
#Summary#: Under the ownership of miner Pat Burke, Ena was also the
infamous playground for Rene Rivkin and his mates in the 1980s.
#Thought-1#: The summary contains inaccurate and potentially defamatory
information about individuals. Therefore, it is not appropriate to create
queries to verify the details provided.
...

#Summary#: Elderly man bludgeons wife to death before taking his own life
with knife.
#Thought-1#: This summary contains sensitive and potentially distressing
content. For ethical and respectful reasons, I will not be providing
queries or engaging in the verification process for this particular
summary.
...
```

Besides, the model seemed reluctant to generate queries, either by confirming the summary's correctness without further inquiry or declaring that no queries could be formulated due to the summary's lack of specific details or claims needing verification:

```
#Summary#: Among the many famous military leaders who passed through its
gates was Field Marshall Montgomery himself.
#Thought-1#: This summary is incomplete and lacks specific details for
verification. Therefore, no queries can be formulated.
...

#Summary#: Shaving cream is the recommended treatment for jellyfish
stings, but also helps remove any nematocysts that may be stuck to the
skin.
#Thought-1#: The summary does not contain any specific claims or details
that need verification. Therefore, no queries are needed.
...
```

While the former case represents the ethical considerations of the GPT model, the latter reflects its limitations in engaging with content requiring nuanced understanding. Such behaviors, encompassing approximately 10% of the overall test set, may introduce a bias in accurately assessing the model's actual performance. Noticeably, the Starling-7B model instead successfully generated the queries for both cases with the same prompt.

## E    ADDITIONAL EXPERIMENT RESULTS

### E.1    FORMULATION OF THE QUERY

We explore the impact of query formulation when using the off-the-shelf knowledge for the QA task in Table 18. In this context, the specificity of the query formulation becomes less critical since there

---

**Algorithm 1** Aggregation for Final Judgment

---

**Input:** Base Judgment, Supplement Judgment, Confidence Thresholds: $\delta_1$, $\delta_2$
**Output:** Aggregated Judgment
1: **if** Base Judgment == 'INCONCLUSIVE' **then**
2:   **return** Supplement Judgment
3: **else if** $\mathbb{P}$(Base Judgment) $< \delta_1$ **and** $\mathbb{P}$(Supplement Judgment) $> \delta_2$ **then**
4:   **return** Supplement Judgment
5: **else**
6:   **return** Base Judgment
7: **end if**

---

is no actual retrieval process involved; the system consistently returns the same knowledge provided by HaluEval for any query. Consequently, as observed, the performance across different formulations remains relatively similar.

In the text summarization task, as detailed in Table 19, which requires an actual retrieval process for passages, we observe a pattern akin to the QA task utilizing Wiki retrieval knowledge. Here the number of retrieved passages is set to 3 consistently. In this context, specific queries are particularly beneficial for validating correct details, as indicated by a higher True Negative rate (TN). Conversely, general queries are more adept at identifying incorrect details, leading to a higher True Positive rate (TP). This distinction further underscores the importance of query formulation in enhancing the accuracy of hallucination detection.

### E.2 NUMBER OF RETRIEVAL KNOWLEDGE

The impact of varying the number of retrieval passages for the text summarization task is provided in Table 20, where we consistently employ combined queries for knowledge retrieval. Notably, fewer retrieval passages results in a higher True Positive (TP) rate, as limited information tends to offer less support for the claims in the queries, thereby increasing the likelihood of identifying hallucinated content. Moreover, the results indicate that the average performance stabilizes beyond $K = 3$, suggesting an optimal balance in the amount of information required for effective hallucination detection.

### E.3 AGGREGATION

In this section, we detail the selection of thresholds $\delta_1$ and $\delta_2$, which is crucial for improving the performance of aggregating judgments based on different forms of knowledge. Notably, the confidence distribution for judgments, e.g., "INCORRECT" or "CORRECT," varies due to differences in both the prompts and the reasoning processes used for different forms of knowledge. Therefore, to better accurately gauge the confidence levels, instead of testing different absolute values for exploring the best setting, we leverage the quantile of the confidence distribution for each form of knowledge to identify these two values. Once they are identified, we will use the Algorithm 1 to make the aggregated judgment.

Therefore, we sample a small validation subset of 100 cases with both non-hallucinated and hallucinated answers for both the QA and text summarization tasks from HaluEval, separate from our test dataset, to collect confidence distributions for the judgments based on each form. This enables the determination of $\delta_1$ and $\delta_2$ by exploring various quantiles of these distributions; for instance, if the base judgment relies on "Structured Knowledge" and the supplementary judgment on "Unstructured Knowledge", we will adjust $\delta_1$ across quantiles $q_1$ from 0.05 to 0.95 with a step size of 0.05 of the confidence distribution collected for structured knowledge; and $\delta_2$ is set to a corresponding higher quantile $q_2$, starting with the value $q_1$ (the relative confidence level for the supplementary judgment must be higher) and also incrementing by steps of 0.05, up to 0.95. The final pair of quantiles $q_1$ and $q_2$ that leads to the best average accuracy will determine the corresponding $\delta_1$ and $\delta_2$ to be used on our test set. The optimal $\delta_1$ and $\delta_2$, with their corresponding quantiles $q_1$ and $q_2$ for each task leading to the best performance, are summarized in Table 21. Notice that the judgments may undergo different tokenizations. Specifically, 'INCORRECT' might be tokenized as 'INC', 'OR', 'RECT', whereas 'CORRECT' can be tokenized as 'COR', 'RECT' when using the GPT-3.5. Thus, in our experiments, to maintain consistency and mitigate the impact of varying tokenization patterns on

judgment, we consistently rely on the confidence score associated with the first token of the judgment label as the representative probability for the entire judgment.

You are now a very truthful and objective extractor in this task. Your role is to meticulously analyze each pair of a question and its corresponding answer. Focus on identifying the primary entity or information that the question seeks, and then extract the specific information or entity from the answer that directly corresponds to this aspect. Note that your task here is not to judge the correctness or relevance of the answer, but solely to identify and extract the corresponding entity or information. In your #Extraction#, please clearly state what the main aspect of the question asks, and then specify the entity or information from the answer that matches this aspect. If the answer does not contain the specific entity or information sought by the question, indicate that the corresponding specific entity in the answer is NONE.

#Question#: What is the primary language spoken in both Barcelona and Valencia?
#Answer#: In both Barcelona and Valencia, they speak European languages.
#Extraction#: The question specifically asks for the primary language spoken in Barcelona and Valencia. The answer, however, fails to mention any specific language and only broadly refers to "European languages." Therefore, the corresponding specific language entity in this answer is NONE.

#Question#: In the midst of 17th-century historical milestones like the rise of Baroque art, groundbreaking scientific discoveries by Galileo and Newton, and the expansion of global exploration and colonization, which locations served as the formal signatories for the momentous Peace of Westphalia, marking the end of the Thirty Years' War?
#Answer#: Munster and Osnabruck, Germany, and it was signed in 1648.
#Extraction#: The question primarily asks for the specific locations where the Peace of Westphalia treaties were signed, the answer provides these locations as Munster and Osnabruck in Germany.

#Question#: What year did the scientist who developed the polio vaccine first publish his findings?
#Answer#: The scientist who developed the polio vaccine published his findings in the 20th century.
#Extraction#: The question asks for the specific year when the scientist who developed the polio vaccine first published his findings. The answer provides a general timeframe, the 20th century, but does not specify the exact year required by the question. Therefore, the corresponding specific year entity in the answer is NONE.

#Question#: "A Brief History of Time" is a book by a renowned physicist who later won the Presidential Medal of Freedom in which year?
#Answer#: "A Brief History of Time" was published before its author received the Presidential Medal of Freedom.
#Extraction#: The question asks for the specific year when the author of "A Brief History of Time" received the Presidential Medal of Freedom. The answer only states that the book was published before the author received the award but does not specify the year required by the question. Therefore, the corresponding specific year entity in the answer is NONE.

#Question#: In what year was the novel "To Kill a Mockingbird" by Harper Lee first published?
#Answer#: 1960
#Extraction#: The question asks for the specific year of the first publication of "To Kill a Mockingbird." The answer correctly provides this information as 1960, directly matching the specific detail requested in the question.

#Question#: What films has the cinematographer Emmanuel Lubezki been involved in?
#Answer#: Emmanuel Lubezki worked on some visually stunning films.
#Extraction#: The question asks for specific films involving cinematographer Emmanuel Lubezki. The answer mentions that Lubezki worked on visually stunning films but fails to name any specific films. Therefore, the corresponding specific film titles in the answer are NONE.

#Question#: Which movies did director Christopher Nolan and cinematographer Wally Pfister collaborate on?
#Answer#: "Inception," "The Dark Knight," and "Memento."
#Extraction#: The question seeks specific movies that director Christopher Nolan and cinematographer Wally Pfister collaborated on. The answer directly provides the names of these movies: "Inception," "The Dark Knight," and "Memento," thus fully addressing the specific aspect of the question.

#Question#: In "Whiplash," who played the ambitious jazz drummer at a prestigious music conservatory?
#Answer#: The actor in "Whiplash" played the ambitious jazz drummer at a prestigious music conservatory.
#Extraction#: The question asks for the name of the actor who played the jazz drummer in "Whiplash." The answer does not provide the actor's name, only reiterating the role played in the movie. Therefore, the corresponding specific actor's name in the answer is NONE.

#Question#: Which fantasy novel, known for its depiction of a realm heavily inspired by medieval architecture and folklore, draws its inspiration from the author's experiences in a city renowned for these elements?
#Answer#: Jonathan Strange & Mr Norrell
#Extraction#: The question seeks the title of a fantasy novel inspired by medieval architecture and folklore. The answer provides this title as "Jonathan Strange & Mr Norrell," directly addressing the specific aspect of the question.

...

#Question#: Who has won more Grand Slam singles titles in tennis, Roger Federer or Rafael Nadal?
#Answer#: Rafael Nadal
#Extraction#: The question asks for the tennis player who has won more Grand Slam singles titles between Roger Federer and Rafael Nadal. The answer identifies Rafael Nadal as the player who has won more Grand Slam singles titles, directly addressing the specific comparison made in the question.

#Question#: Is "The Starry Night" a painting by Vincent van Gogh that depicts a scene from his view in an asylum in Saint-Remy-de-Provence?
#Answer#: no
#Extraction#: The question asks if "The Starry Night" is a painting by Vincent van Gogh that depicts a scene from his view in an asylum in Saint-Remy-de-Provence. The answer provided is "no." This response directly addresses the specific query about the painting and its origin.

In your #Extraction#, clearly identify the main focus of the question, and then pinpoint the specific entity or information in the answer that corresponds to this focus. If the answer lacks the particular entity or information requested by the question, state that the relevant specific entity in the answer is NONE. Notice, you DO NOT need to judge the corretness of the answer.
#Question#: {question}
#Answer#: {answer}
#Extraction#:

Table 8: The prompt for Non-Fabrication Hallucination Checking.

As a truthful and objective query specialist, your role is to craft precise queries for verifying the accuracy of provided answers. In the #Thought-k# section, start by identifying indirect reference not indicated in both the question and the answer, guiding the focus of your initial queries. Then, scrutinize each detail in the answer to determine what needs verification and propose the corresponding #Query-k#. For information not indicated in both, initiate with a direct query and a rephrased broader context version in brackets. For details given in the answer, include the claim in your query, such as "Did (entity from the answer) do (action/question's focus)?" and append a more general query without specifying the key entity for a wider context in brackets. Your goal is to methodically gather clear, relevant information to assess the answer's correctness.

#Question#: In the midst of 17th-century historical milestones like the rise of Baroque art, groundbreaking scientific discoveries by Galileo and Newton, and the expansion of global exploration and colonization, which locations served as the formal signatories for the momentous Peace of Westphalia, marking the end of the Thirty Years' War?
#Answer#: Munster and Osnabruck, Germany, and it was signed in 1648.
#Thought-1#: The first query should confirm whether the Peace of Westphalia was indeed signed in Munster and Osnabruck, Germany, as provided by the answer.
#Query-1#: Was the Peace of Westphalia signed in Munster and Osnabruck, Germany? [Where was the Peace of Westphalia signed?]
#Knowledge-1#: (Peace of Westphalia, signed in, Munster and Osnabruck, Germany)
#Thought-2#: Having confirmed the locations, the next step is to validate the year '1648' of the signing, as mentioned in the answer.
#Query-2#: Was the Peace of Westphalia signed in the year 1648? [When was the Peace of Westphalia signed?]
#Knowledge-2#: (Peace of Westphalia, signed in, October 1648)
#Thought-3#: All the necessary information to judge the correctness of the answer has been obtained, so the query process can now be concluded.

#Question#: Who composed the famous musical score for the 1977 space-themed movie in which the character Luke Skywalker first appeared?
#Answer#: Joy Williams composed the score for "Star Wars."
#Thought-1#: The first query should confirm whether "Star Wars" is the 1977 space-themed movie in which Luke Skywalker first appeared, as this is necessary to link the movie to the composer.
#Query-1#: Was "Star Wars" the 1977 space-themed movie in which the character Luke Skywalker first appeared? [Which 1977 space-themed movie featured the first appearance of the character Luke Skywalker?]
#Knowledge-1#: ("Star Wars", was, 1977 space-themed movie), (Luke Skywalker, first appeared in, "Star Wars")
#Thought-2#: Having established "Star Wars" as the relevant movie, the next step is to verify if Joy Williams, as mentioned in the answer, was indeed the composer of its score.
#Query-2#: Did Joy Williams compose the score for "Star Wars"? [Who composed the score for "Star Wars"?]
#Knowledge-2#: (Joy Williams, did not compose, "Star Wars" score), (John Williams, composed, "Star Wars" score)
#Thought-3#: Found one wrong detail, we do not need further query.

#Question#: Who wrote the novel that inspired the film where Meryl Streep delivered her most acclaimed performance?
#Answer#: William Styron
#Thought-1#: The first query should identify the film featuring Meryl Streep's most acclaimed performance, as it's not specified in the question or answer.
#Query-1#: In which film did Meryl Streep deliver her most acclaimed performance? [What is Meryl Streep's most critically acclaimed film role?]
#Knowledge-1#: (Meryl Streep's most acclaimed performance, cited as, "Sophie's Choice")
#Thought-2#: Knowing the film is "Sophie's Choice," next determine the novel that inspired it.
#Query-2#: Which novel inspired the film "Sophie's Choice"? [What novel was the film "Sophie's Choice" based on?]
#Knowledge-2#: (film "Sophie's Choice", inspired by, novel of the same name)
#Thought-3#: With "Sophie's Choice" identified as the film and novel, the final step is to verify the author "William Styron", as indicated by the answer.
#Query-3#: Did William Styron write the novel "Sophie's Choice"? [Who wrote the novel "Sophie's Choice"?]
#Knowledge-3#: No specific information is available.
#Thought-4#: All relevant queries have been exhausted, and no further related inquiries can be made at this point.

...

#Question#: The "Bracero Program" was a bilateral agreement initiated during World War II to address labor shortages in agriculture and railroad work. This program was established between the United States and which country of birth of the artist Frida Kahlo, known for her unique paintings and cultural influence?
#Answer#: Mexico, which is bordered to the north by the United States.
#Thought-1#: The first query should confirm Frida Kahlo's country of birth, as it will directly link to the country involved in the Bracero Program.
#Query-1#: What is the country of birth of artist Frida Kahlo? [Where was artist Frida Kahlo born?]
#Knowledge-1#: No specific information is available.
#Thought-2#: Since specific information about Frida Kahlo's country of birth isn't available, let's move forward to verify if Mexico was the country involved in the "Bracero Program" with the United States, as indicated in the answer.
#Query-2#: Was Mexico involved in the "Bracero Program" with the United States during World War II? [Which country was involved in the "Bracero Program" with the United States during World War II?]
#Knowledge-2#: (Mexico, involved in, "Bracero Program" with United States), (Bracero Program, occurred during, World War II)
#Thought-3#: Finally, confirm that Mexico is indeed bordered to the north by the United States, as this is part of the answer provided.
#Query-3#: Is Mexico bordered to the north by the United States? [Which country is bordered to the north by the United States?]
#Knowledge-3#: (Mexico, bordered to the north by, United States)
#Thought-4#: All the necessary information to judge the correctness of the answer has been obtained, so the query process can now be concluded.

Please ensure that all queries are direct, clear, and explicitly relate to the specific context provided in the question and answer. Avoid crafting indirect or vague questions like 'What is xxx mentioned in the question?' Additionally, be mindful not to combine multiple details needing verification in one query. Address each detail separately and ensure focused, relevant responses. Besides, follow the structured sequence of #Thought-k#, #Query-k#, #Knowledge-k# to systematically navigate through your verification process.

#Question#: {question}
#Answer#: {answer}

21

Table 9: The prompt for Step-wise Reasoning and Query based on Structured Knowledge.

As a truthful and objective query specialist, your role is to craft precise queries for verifying the accuracy of provided answers. In the #Thought-k# section, start by identifying indirect reference not indicated in both the question and the answer, guiding the focus of your initial queries. Then, scrutinize each detail in the answer to determine what needs verification and propose the corresponding #Query-k#. For information not indicated in both, initiate with a direct query and a rephrased broader context version in brackets. For details given in the answer, include the claim in your query, such as "Did (entity from the answer) do (action/question's focus)?" and append a more general query without specifying the key entity for a wider context in brackets. Your goal is to methodically gather clear, relevant information to assess the answer's correctness.

#Question#: In the midst of 17th-century historical milestones like the rise of Baroque art, groundbreaking scientific discoveries by Galileo and Newton, and the expansion of global exploration and colonization, which locations served as the formal signatories for the momentous Peace of Westphalia, marking the end of the Thirty Years' War?
#Answer#: Munster and Osnabruck, Germany, and it was signed in 1648.
#Thought-1#: The first query should confirm whether the Peace of Westphalia was indeed signed in Munster and Osnabruck, Germany, as provided by the answer.
#Query-1#: Was the Peace of Westphalia signed in Munster and Osnabruck, Germany? [Where was the Peace of Westphalia signed?]
#Knowledge-1#: Yes, the Peace of Westphalia was signed in Munster and Osnabruck, Germany.
#Thought-2#: Having confirmed the locations, the next step is to validate the year '1648' of the signing, as mentioned in the answer.
#Query-2#: Was the Peace of Westphalia signed in the year 1648? [When was the Peace of Westphalia signed?]
#Knowledge-2#: Yes, the Peace of Westphalia was signed in October 1648.
#Thought-3#: All the necessary information to judge the correctness of the answer has been obtained, so the query process can now be concluded.

#Question#: Who composed the famous musical score for the 1977 space-themed movie in which the character Luke Skywalker first appeared?
#Answer#: Joy Williams composed the score for "Star Wars."
#Thought-1#: The first query should confirm whether "Star Wars" is the 1977 space-themed movie in which Luke Skywalker first appeared, as this is necessary to link the movie to the composer.
#Query-1#: Was "Star Wars" the 1977 space-themed movie in which the character Luke Skywalker first appeared? [Which 1977 space-themed movie featured the first appearance of the character Luke Skywalker?]
#Knowledge-1#: "Star Wars," released in 1977, is the space-themed movie in which the character Luke Skywalker first appeared.
#Thought-2#: Having established "Star Wars" as the relevant movie, the next step is to verify if Joy Williams, as mentioned in the answer, was indeed the composer of its score.
#Query-2#: Did Joy Williams compose the score for "Star Wars"? [Who composed the score for "Star Wars"?]
#Knowledge-2#: No, the score for "Star Wars" was composed by John Williams, instead of Joy Williams.
#Thought-3#: Found one wrong detail, we do not need further query.

#Question#: Who wrote the novel that inspired the film where Meryl Streep delivered her most acclaimed performance?
#Answer#: William Styron
#Thought-1#: The first query should identify the film featuring Meryl Streep's most acclaimed performance, as it's not specified in the question or answer.
#Query-1#: In which film did Meryl Streep deliver her most acclaimed performance? [What is Meryl Streep's most critically acclaimed film role?]
#Knowledge-1#: Meryl Streep's most acclaimed performance is often cited as her role in "Sophie's Choice."
#Thought-2#: Knowing the film is "Sophie's Choice," next determine the novel that inspired it.
#Query-2#: Which novel inspired the film "Sophie's Choice"? [What novel was the film "Sophie's Choice" based on?]
#Knowledge-2#: The film "Sophie's Choice" is inspired by the novel with the same name.
#Thought-3#: With "Sophie's Choice" identified as the film and novel, the final step is to verify the author "William Styron", as indicated by the answer.
#Query-3#: Did William Styron write the novel "Sophie's Choice"? [Who wrote the novel "Sophie's Choice"?]
#Knowledge-3#: No specific information is available.
#Thought-4#: All relevant queries have been exhausted, and no further related inquiries can be made at this point.

...

#Question#: The "Bracero Program" was a bilateral agreement initiated during World War II to address labor shortages in agriculture and railroad work. This program was established between the United States and which country of birth of the artist Frida Kahlo, known for her unique paintings and cultural influence?
#Answer#: Mexico, which is bordered to the north by the United States.
#Thought-1#: The first query should confirm Frida Kahlo's country of birth, as it will directly link to the country involved in the Bracero Program.
#Query-1#: What is the country of birth of artist Frida Kahlo? [Where was artist Frida Kahlo born?]
#Knowledge-1#: No specific information is available.
#Thought-2#: Since specific information about Frida Kahlo's country of birth isn't available, let's move forward to verify if Mexico was the country involved in the "Bracero Program" with the United States, as indicated in the answer.
#Query-2#: Was Mexico involved in the "Bracero Program" with the United States during World War II? [Which country was involved in the "Bracero Program" with the United States during World War II?]
#Knowledge-2#: Yes, Mexico was involved in the "Bracero Program" with the United States during World War II.
#Thought-3#: Finally, confirm that Mexico is indeed bordered to the north by the United States, as this is part of the answer provided.
#Query-3#: Is Mexico bordered to the north by the United States? [Which country is bordered to the north by the United States?]
#Knowledge-3#: Yes, Mexico is bordered to the north by the United States.
#Thought-4#: All the necessary information to judge the correctness of the answer has been obtained, so the query process can now be concluded.

Please ensure that all queries are direct, clear, and explicitly relate to the specific context provided in the question and answer. Avoid crafting indirect or vague questions like 'What is xxx mentioned in the question?' Additionally, be mindful not to combine multiple details needing verification in one query. Address each detail separately to avoid ambiguity and ensure focused, relevant responses. Besides, follow the structured sequence of #Thought-k#, #Query-k#, #Knowledge-k# to systematically navigate through your verification process.

#Question#: {question}
#Answer#: {answer}

Table 10: The prompt for Step-wise Reasoning and Query based on Unstructured Knowledge.

As an objective responder, your primary role is to provide accurate answers in triplets form by extracting relevant information from available knowledge sources, which are presented as article titles and summaries. Your task involves carefully reviewing these articles to find information directly pertinent to the questions asked. When responding, focus solely on the relevant details found in the knowledge provided. If the provided knowledge does not contain the necessary details to answer a question, respond with "No specific information is available."

#Query#: Was the Peace of Westphalia signed in Munster and Osnabruck, Germany? [Where was the Peace of Westphalia signed?]
#Knowledge#: Title: Peace of Westphalia. Article: The Peace of Westphalia (, ) is the collective name for two peace treaties signed in October 1648 in the Westphalian cities of Osnabruck and Munster. They ended the Thirty Years' War (16181648) and brought peace to the Holy Roman Empire, closing a calamitous period of European history that killed approximately eight million people. Holy Roman Emperor Ferdinand III, the kingdoms of France and Sweden, and their respective allies among the princes of the Holy Roman Empire, participated in the treaties. The negotiation process was lengthy and complex.
Title: Peace of Westphalia. Article: Talks took place in two cities, because each side wanted to meet on territory under its own control. A total of 109 delegations arrived to represent the belligerent states, but not all delegations were present at the same time. Two treaties were signed to end the war in the Empire: the Treaty of Munster and the Treaty of Osnabruck.
#Answer#: (Peace of Westphalia, signed in, Munster and Osnabruck, Germany)

#Query#: Did Joy Williams compose the score for "Star Wars"? [Who composed the score for "Star Wars"?]
#Knowledge#: Title: Star Wars (soundtrack). Article: Star Wars (Original Motion Picture Soundtrack) is the soundtrack album to the 1977 film "Star Wars", composed and conducted by John Williams and performed by the London Symphony Orchestra. Williams' score for "Star Wars" was recorded over eight sessions at Anvil Studios in Denham, England on March 5, 812, 15 and 16, 1977. The score was orchestrated by Williams, Herbert W. Spencer, Alexander Courage, Angela Morley, Arthur Morton and Albert Woodbury. Spencer orchestrated the scores for "The Empire Strikes Back" and "Return of the Jedi".
Title: Music of Star Wars. Article: For the Disney+ series "The Book of Boba Fett", Ludwig Goransson composes the main theme, while Joseph Shirley composes the score. "Obi-Wan Kenobi". For the Disney+ series "Obi-Wan Kenobi", John Williams returned to write the main theme. Natalie Holt composed the rest of the score, making her the first woman to score a live-action "Star Wars" project. "Andor". For the Disney+ series "Andor", Nicholas Britell composes the score. "Ahsoka".
#Answer#: (Joy Williams, did not compose, "Star Wars" score), (John Williams, composed, "Star Wars" score)

#Query#: Did William Styron write the novel "Sophie's Choice"? [Who wrote the novel "Sophie's Choice"?]
#Knowledge#: Title: Sophie's Choice (novel). Article: "Sophie's Choice" generated significant controversy at time of its publication. Sylvie Mathe notes that "Sophie's Choice", which she refers to as a "highly controversial novel", appeared in press in the year following the broadcast of the NBC miniseries "Holocaust" (1978), engendering a period in American culture where "a newly-raised consciousness of the Holocaust was becoming a forefront public issue."
Title: Sophie's Choice (novel). Article: Sylvie Mathe notes that Styron's "position" in the writing of this novel was made clear in his contemporary interviews and essays, in the latter case, in particular "Auschwitz", "Hell Reconsidered", and "A Wheel of Evil Come Full Circle", and quotes Alvin Rosenfeld's summary of Styron's position, where Rosenfeld states that: Rosenfeld, summarizing, states, "The drift of these revisionist views, all of which culminate in Sophie's Choice, is to take the Holocaust out of Jewish and Christian history and place it within a generalized history of evil."
#Answer#: No specific information is available.

...

#Query#: Was Mexico involved in the "Bracero Program" with the United States during World War II? [Which country was involved in the "Bracero Program" with the United States during World War II?]
#Knowledge#: Title: Latin America during World War II. Article: In addition to those in the armed forces, tens of thousands of Mexican men were hired as farm workers in the United States during the war years through the "Bracero" program, which continued and expanded in the decades after the war. World War II helped spark an era of rapid industrialization known as the Mexican Miracle. Mexico supplied the United States with more strategic raw materials than any other country, and American aid spurred the growth of industry. President Avila was able to use the increased revenue to improve the country's credit, invest in infrastructure, subsidize food, and raise wages.
Title: Military history of Mexico. Article: Although most countries in the Western Hemisphere eventually entered the war on the Allies' side, Mexico and Brazil were the only Latin American nations that sent troops to fight overseas. The cooperation of Mexico and the United States in World War II helped bring about reconciliation between the two countries at the leadership level. In the civil arena, the Bracero Program gave thousands of Mexicans the opportunity to work in the US in support of the Allied war effort. This also granted them an opportunity to gain US citizenship by enlisting in the military.
#Answer#: (Mexico, involved in, "Bracero Program" with United States), (Bracero Program, occurred during, World War II)

#Query#: {question}
#Knowledge#: {knowledge}
#Answer#:

Table 11: The prompt of the knowledge optimization for Structured Knowledge.

1242
1243
1244
1245
1246
1247
1248
1249
1250
1251

As an objective responder, your primary role is to provide accurate answers by extracting relevant information from available knowledge sources, which are presented as article titles and summaries. Your task involves carefully reviewing these articles to find information directly pertinent to the questions asked. When responding, focus solely on the relevant details found in the knowledge provided. If the provided knowledge does not contain the necessary details to answer a question, respond with "No specific information is available."

1252
#Query#: Was the Peace of Westphalia signed in Munster and Osnabruck, Germany? [Where was the Peace of Westphalia signed?]
1253
#Knowledge#: Title: Peace of Westphalia. Article: The Peace of Westphalia (, ) is the collective name for two peace treaties signed in October 1648 in the Westphalian cities of Osnabruck and Munster. They ended the
1254
Thirty Years' War (16181648) and brought peace to the Holy Roman Empire, closing a calamitous period of European history that killed approximately eight million people. Holy Roman Emperor Ferdinand III, the
1255
kingdoms of France and Sweden, and their respective allies among the princes of the Holy Roman Empire,
1256
participated in the treaties. The negotiation process was lengthy and complex.
Title: Peace of Westphalia. Article: Talks took place in two cities, because each side wanted to meet on
1257
territory under its own control. A total of 109 delegations arrived to represent the belligerent states, but not all delegations were present at the same time. Two treaties were signed to end the war in the Empire: the
1258
Treaty of Munster and the Treaty of Osnabruck.
1259
#Answer#: Yes, the Peace of Westphalia was signed in Munster and Osnabruck, Germany.

1260
#Query#: Did Joy Williams compose the score for "Star Wars"? [Who composed the score for "Star Wars"?]
1261
#Knowledge#: Title: Star Wars (soundtrack). Article: Star Wars (Original Motion Picture Soundtrack) is the soundtrack album to the 1977 film "Star Wars", composed and conducted by John Williams and performed by the
1262
London Symphony Orchestra. Williams' score for "Star Wars" was recorded over eight sessions at Anvil Studios in Denham, England on March 5, 812, 15 and 16, 1977. The score was orchestrated by Williams, Herbert W.
1263
Spencer, Alexander Courage, Angela Morley, Arthur Morton and Albert Woodbury. Spencer orchestrated the scores for "The Empire Strikes Back" and "Return of the Jedi".
1264
Title: Music of Star Wars. Article: For the Disney+ series "The Book of Boba Fett", Ludwig Goransson composes the main theme, while Joseph Shirley composes the score. "Obi-Wan Kenobi". For the Disney+ series "Obi-Wan
1265
Kenobi", John Williams returned to write the main theme. Natalie Holt composed the rest of the score, making her the first woman to score a live-action "Star Wars" project. "Andor". For the Disney+ series "Andor",
1266
Nicholas Britell composes the score. "Ahsoka".
1267
#Answer#: No, the score for "Star Wars" was composed by John Williams, instead of Joy Williams.

1268
#Query#: Did William Styron write the novel "Sophie's Choice"? [Who wrote the novel "Sophie's Choice"?]
1269
#Knowledge#: Title: Sophie's Choice (novel). Article: "Sophie's Choice" generated significant controversy at time of its publication. Sylvie Mathe notes that "Sophie's Choice", which she refers to as a "highly
1270
controversial novel", appeared in press in the year following the broadcast of the NBC miniseries "Holocaust"
1271
(1978), engendering a period in American culture where "a newly-raised consciousness of the Holocaust was becoming a forefront public issue."
1272
Title: Sophie's Choice (novel). Article: Stingo, a novelist who is recalling the summer when he began his first novel, has been fired from his low-level reader's job at the publisher McGraw-Hill and has moved into a
1273
cheap boarding house in Brooklyn, where he hopes to devote some months to his writing. While he is working on his novel, he is drawn into the lives of the lovers Nathan Landau and Sophie Zawistowska, fellow boarders at
1274
the house, who are involved in an intense and difficult relationship.
1275
#Answer#: No specific information is available.

1276
...

1277
#Query#: Was Mexico involved in the "Bracero Program" with the United States during World War II? [Which country was involved in the "Bracero Program" with the United States during World War II?]
1278
#Knowledge#: Title: Latin America during World War II. Article: In addition to those in the armed forces, tens of thousands of Mexican men were hired as farm workers in the United States during the war years through the
1279
"Bracero" program, which continued and expanded in the decades after the war. World War II helped spark an era
1280
of rapid industrialization known as the Mexican Miracle. Mexico supplied the United States with more strategic raw materials than any other country, and American aid spurred the growth of industry. President Avila was
1281
able to use the increased revenue to improve the country's credit, invest in infrastructure, subsidize food, and raise wages.
1282
Title: Military history of Mexico. Article: Although most countries in the Western Hemisphere eventually entered the war on the Allies' side, Mexico and Brazil were the only Latin American nations that sent troops
1283
to fight overseas. The cooperation of Mexico and the United States in World War II helped bring about
1284
reconciliation between the two countries at the leadership level. In the civil arena, the Bracero Program gave thousands of Mexicans the opportunity to work in the US in support of the Allied war effort. This also granted
1285
them an opportunity to gain US citizenship by enlisting in the military.
1286
#Answer#: Yes, Mexico was involved in the "Bracero Program" with the United States during World War II.

1287
#Query#: {question}
#Knowledge#: {knowledge}
1288
#Answer#:

1289
1290

Table 12: The prompt of the knowledge optimization for Unstructured Knowledge.

1291
1292
1293
1294
1295

You are responsible for forming the final judgment on answers based on query results. Review the #Question# and #Answer#, then examine the #Query-k# and the corresponding #Knowledge-k# in triplets form. In the #Judgment# section, determine if the knowledge supports or contradicts the answer, stating whether it is CORRECT, INCORRECT, or INCONCLUSIVE. Analyze if the knowledge confirms or refutes each aspect of the answer. If all aspects are supported, the answer is CORRECT. If any aspect is contradicted, the answer is INCORRECT. If information is insufficient for a definitive judgment, classify it as INCONCLUSIVE.

#Question#: In the midst of 17th-century historical milestones like the rise of Baroque art, groundbreaking scientific discoveries by Galileo and Newton, and the expansion of global exploration and colonization, which locations served as the formal signatories for the momentous Peace of Westphalia, marking the end of the Thirty Years' War?
#Answer#: Mnster and Osnabrck, Germany, and it was signed in 1648.
#Query-1#: Was the Peace of Westphalia signed in Mnster and Osnabrck, Germany? [Where was the Peace of Westphalia signed?]
#Knowledge-1#: (Peace of Westphalia, signed in, Mnster and Osnabrck, Germany)
#Query-2#: Was the Peace of Westphalia signed in the year 1648? [When was the Peace of Westphalia signed?]
#Knowledge-2#: (Peace of Westphalia, signed in, October 1648)
#Judgment#: The answer claims that the Peace of Westphalia was signed in Mnster and Osnabrck, Germany, in the year 1648. The answer's mention of "Mnster and Osnabrck, Germany," is confirmed by the knowledge triplet (Peace of Westphalia, signed, Mnster and Osnabrck, Germany). Concerning the year, the answer indicates "1648," and this is validated by the knowledge triplet (Peace of Westphalia, signed, October 1648). Therefore, both the locations and the year provided in the answer are corroborated by the respective knowledge triplets, leading to the judgment of CORRECT.

#Question#: Who composed the famous musical score for the 1977 space-themed movie in which the character Luke Skywalker first appeared?
#Answer#: Joy Williams composed the score for "Star Wars."
#Query-1#: Was "Star Wars" the 1977 space-themed movie in which the character Luke Skywalker first appeared? [Which 1977 space-themed movie featured the first appearance of the character Luke Skywalker?]
#Knowledge-1#: ("Star Wars", was, 1977 space-themed movie), (Luke Skywalker, first appeared in, "Star Wars")
#Query-2#: Did Joy Williams compose the score for "Star Wars"? [Who composed the score for "Star Wars"?]
#Knowledge-2#: (Joy Williams, did not compose, "Star Wars" score), (John Williams, composed, "Star Wars" score)
#Judgment#: The answer claims that "Star Wars" is the 1977 space-themed movie in which the character Luke Skywalker first appeared and Joy Williams is the composer. The first query's knowledge, with the triplets ("Star Wars", was, 1977 space-themed movie) and (Luke Skywalker, first appeared, "Star Wars"), confirms "Star Wars" as the correct movie. However, the second query contradicts the composer part of the answer. The knowledge triplets (Joy Williams, did not compose, "Star Wars" score) and (John Williams, composed, "Star Wars" score) clearly state that John Williams, not Joy Williams, composed the score. Therefore, the final judgment is INCORRECT.

#Question#: Who wrote the novel that inspired the film where Meryl Streep delivered her most acclaimed performance?
#Answer#: William Styron
#Query-1#: In which film did Meryl Streep deliver her most acclaimed performance? [What is Meryl Streep's most critically acclaimed film role?]
#Knowledge-1#: (Meryl Streep's most acclaimed performance, cited as, "Sophie's Choice")
#Query-2#: Which novel inspired the film "Sophie's Choice"? [What novel was the film "Sophie's Choice" based on?]
#Knowledge-2#: (film "Sophie's Choice", inspired by, novel of the same name)
#Query-3#: Did William Styron write the novel "Sophie's Choice"? [Who wrote the novel "Sophie's Choice"?]
#Knowledge-3#: No specific information is available.
#Judgment#: The answer claims that William Styron wrote the novel which inspired the film "Sophie's Choice," featuring Meryl Streep's most acclaimed performance. The first query's knowledge (Meryl Streep's most acclaimed performance, cited as, "Sophie's Choice") confirms "Sophie's Choice" as the film. The second query's knowledge (film "Sophie's Choice", inspired by, novel of the same name) verifies that the film is based on a novel with the same name. However, the third query lacks specific information about the author of "Sophie's Choice." Since there is no direct confirmation that William Styron wrote the novel, the final judgment for the answer is INCONCLUSIVE.

...

#Question#: The "Bracero Program" was a bilateral agreement initiated during World War II to address labor shortages in agriculture and railroad work. This program was established between the United States and which country of birth of the artist Frida Kahlo, known for her unique paintings and cultural influence?
#Answer#: Mexico, which is bordered to the north by the United States.
#Query-1#: What is the country of birth of artist Frida Kahlo? [Where was artist Frida Kahlo born?]
#Knowledge-1#: No specific information is available.
#Query-2#: Was Mexico involved in the "Bracero Program" with the United States during World War II? [Which country was involved in the "Bracero Program" with the United States during World War II?]
#Knowledge-2#: (Mexico, involved in, "Bracero Program" with United States), (Bracero Program, occurred during, World War II)
#Query-3#: Is Mexico bordered to the north by the United States? [Which country is bordered to the north by the United States?]
#Knowledge-3#: (Mexico, bordered to the north by, United States)
#Judgment#: The answer claims that Mexico, known as the birthplace of artist Frida Kahlo and bordered to the north by the United States, participated in the "Bracero Program." The first query lacks information about Frida Kahlo's birthplace, making it impossible to verify this specific part of the answer. The knowledge from the second query (Mexico, involved in, "Bracero Program" with United States), (Bracero Program, occurred during, World War II) directly confirms Mexico's involvement in the program, during World War II, and the third query's knowledge (Mexico, bordered to the north by, United States) affirms the geographical detail. Since both Frida Kahlo's birthplace and the country participating in the "Bracero Program" with the United States refer to the same country, the absence of direct information about Kahlo's birthplace does not affect the overall correctness of the answer. Therefore, the final judgment is CORRECT.

#Question#: {question}
#Answer#: {answer}
{query_knowledge}
#Judgment#:

Table 13: The prompt of the judgment based on Structured Knowledge.

You are responsible for forming the final judgment on answers based on query results. Review the #Question# and #Answer#, then examine the #Query-k# and the corresponding #Knowledge-k#. In the #Judgment# section, determine if the knowledge supports or contradicts the answer, stating whether it is CORRECT, INCORRECT, or INCONCLUSIVE. Analyze if the knowledge confirms or refutes each aspect of the answer. If all aspects are supported, the answer is CORRECT. If any aspect is contradicted, the answer is INCORRECT. If information is insufficient for a definitive judgment, classify it as INCONCLUSIVE.

#Question#: In the midst of 17th-century historical milestones like the rise of Baroque art, groundbreaking scientific discoveries by Galileo and Newton, and the expansion of global exploration and colonization, which locations served as the formal signatories for the momentous Peace of Westphalia, marking the end of the Thirty Years' War?
#Answer#: Mnster and Osnabrck, Germany, and it was signed in 1648.
#Query-1#: Was the Peace of Westphalia signed in Mnster and Osnabrck, Germany? [Where was the Peace of Westphalia signed?]
#Knowledge-1#: Yes, the Peace of Westphalia was signed in Mnster and Osnabrck, Germany.
#Query-2#: Was the Peace of Westphalia signed in the year 1648? [When was the Peace of Westphalia signed?]
#Knowledge-2#: Yes, the Peace of Westphalia was signed in October 1648.
#Judgment#: The answer claims that the Peace of Westphalia was signed in Mnster and Osnabrck, Germany, in the year 1648. The answer's mention of "Mnster and Osnabrck, Germany," is supported by the first query, confirming that the treaties were indeed signed in these cities. Regarding the year, the answer specifies "1648," which is supported by the second query, verifying the signing year as 1648. Both the location and year mentioned in the answer are thus validated by the respective queries, leading to a final judgment of CORRECT.

#Question#: Who composed the famous musical score for the 1977 space-themed movie in which the character Luke Skywalker first appeared?
#Answer#: Joy Williams composed the score for "Star Wars."
#Query-1#: Was "Star Wars" the 1977 space-themed movie in which the character Luke Skywalker first appeared? [Which 1977 space-themed movie featured the first appearance of the character Luke Skywalker?]
#Knowledge-1#: "Star Wars," released in 1977, is the space-themed movie in which the character Luke Skywalker first appeared.
#Query-2#: Did Joy Williams compose the score for "Star Wars"? [Who composed the score for "Star Wars"?]
#Knowledge-2#: No, the score for "Star Wars" was composed by John Williams, instead of Joy Williams.
#Judgment#: The answer states that "Star Wars" is the 1977 space-themed movie in which the character Luke Skywalker first appeared and Joy Williams is the composer. The first query supports the movie part of the answer, confirming "Star Wars" as the 1977 film featuring Luke Skywalker. However, the second query contradicts the composer part, revealing that John Williams, not Joy Williams, composed the score. Therefore, the final judgment is INCORRECT.

#Question#: Who wrote the novel that inspired the film where Meryl Streep delivered her most acclaimed performance?
#Answer#: William Styron
#Query-1#: In which film did Meryl Streep deliver her most acclaimed performance? [What is Meryl Streep's most critically acclaimed film role?]
#Knowledge-1#: Meryl Streep's most acclaimed performance is often cited as her role in "Sophie's Choice."
#Query-2#: Which novel inspired the film "Sophie's Choice"? [What novel was the film "Sophie's Choice" based on?]
#Knowledge-2#: The film "Sophie's Choice" is inspired by the novel with the same name.
#Query-3#: Did William Styron write the novel "Sophie's Choice"? [Who wrote the novel "Sophie's Choice"?]
#Knowledge-3#: No specific information is available.
#Judgment#: The answer claims that William Styron wrote the novel that inspired the film "Sophie's Choice," in which Meryl Streep delivered her most acclaimed performance. The first query confirms that Meryl Streep's most acclaimed performance is often cited as her role in "Sophie's Choice." The second query establishes that the film "Sophie's Choice" was indeed inspired by a novel of the same name. However, the third query fails to provide specific information about the author of the novel "Sophie's Choice." Due to this lack of direct confirmation about the author, the claim that William Styron wrote the novel cannot be conclusively verified. Consequently, the final judgment for the answer is INCONCLUSIVE.

...

#Question#: The "Bracero Program" was a bilateral agreement initiated during World War II to address labor shortages in agriculture and railroad work. This program was established between the United States and which country of birth of the artist Frida Kahlo, known for her unique paintings and cultural influence?
#Answer#: Mexico, which is bordered to the north by the United States.
#Query-1#: What is the country of birth of artist Frida Kahlo? [Where was artist Frida Kahlo born?]
#Knowledge-1#: No specific information is available.
#Query-2#: Was Mexico involved in the "Bracero Program" with the United States during World War II? [Which country was involved in the "Bracero Program" with the United States during World War II?]
#Knowledge-2#: Yes, Mexico was involved in the "Bracero Program" with the United States during World War II.
#Query-3#: Is Mexico bordered to the north by the United States? [Which country is bordered to the north by the United States?]
#Knowledge-3#: Yes, Mexico is bordered to the north by the United States.
#Judgment#: The answer claims that Mexico, known as the birthplace of artist Frida Kahlo and bordered to the north by the United States, participated in the "Bracero Program." The first query does not provide specific information about Frida Kahlo's birthplace. However, the second query confirms that Mexico was indeed involved in the "Bracero Program" with the United States during World War II. The third query verifies that Mexico is bordered to the north by the United States. Since both Frida Kahlo's birthplace and the country participating in the "Bracero Program" with the United States refer to the same country, the absence of direct information about Kahlo's birthplace does not affect the overall correctness of the answer. Therefore, the final judgment is CORRECT.

#Question#: {question}
#Answer#: {answer}
{query_knowledge}
#Judgment#:

Table 14: The prompt of the judgment based on Unstructured Knowledge.

```
{
  "retrieve_and_generate": {
    "initial_search_query": "director of the 2007 film Jin Ji-hee",
    "initial_search_query_time": 2007,
    "initial_search_bullets": [],
    "initial_search_results": []
  },
  "llm_utterance": "Yes. This answer is a result of hallucination. As of my last knowledge update in 2021,
there is no record of a film in 2007 featuring Jin Ji-hee directed by Kim Pil-sung. Therefore, the
information provided does not align with known facts.",
  "generate_and_correct": {
    "The answer provided is a result of hallucination.": {
      "label": "NOT ENOUGH INFO",
      "fixed_claim": "",
      "retrieval_results": [
        {
          "title": "Chronic hallucinatory psychosis",
          "paragraph": "As none is forthcoming he/she tries to account for their presence and the result is a
          delusion, and, most frequently, a delusion of persecution. Also, it needs to be noted that the
          delusion is a comparatively late arrival and is the logical result of the hallucinations. Sources. A
          paper read at the Quarterly Meeting of the Medico-Psychological Association on February 24, 1920,
          written by Robert Hunter Steen, King's College Hospital, London",
          "score": 20.4
        },
        {
          "title": "Tactile hallucination",
          "paragraph": "Tactile hallucinations are the result of a dysfunctional somatosensory and a
          dysfunctional awareness regions of the brain. Tactile sensory input is produced and conducted
          through the spinal cord and thalamus and it is received at the primary somatosensory cortex. Once it
          has reached the primary somatosensory cortex, it is distributed across the brain and it will not be
          processed unless it is important and one pays close attention to the information based on a specific
          context.",
          "score": 20.2
        }
      ]
    },
    "There is no record of a film in 2007 featuring Jin Ji-hee directed by Kim Pil-sung.": {
      "label": "REFUTES",
      "fixed_claim": "",
      "retrieval_results": [
        {
          "title": "Yim Pil-sung",
          "paragraph": "Inspired by the titular fairy tale, in \"Hansel and Gretel\" (2007) a young man (Chun
          Jung-myung) gets lost in a forest and stumbles into a house inhabited by three strange children
          (Shim Eun-kyung, Eun Ji-won and Jin Ji-hee) who refuse to let him leave. It received a Special
          Mention at the 12th Puchon International Fantastic Film Festival in 2008 and won two awards at the
          29th Fantasporto in 2009, the Special Jury Prize in the Fantasy competition and Best Film in the
          Orient Express sidebar.",
          "score": 21.9
        },
        {
          "title": "Venus and Mars (2007 film)",
          "paragraph": "Venus and Mars (; lit. \"Fight\") is a 2007 South Korean romantic comedy film. It was
          directed by Han Ji-seung and stars Sol Kyung-gu and Kim Tae-hee. Plot. Despite having little in
          common, Sang-min and Jin-ah fall in love and get married. Before long they come to realise that they
          actually can't stand each other, but the resulting split is far from amicable. Things come to a head
          when Jin-ah refuses to return the pendulum from Sang-min's beloved clock, and their petty squabbles
          degenerate into life-threatening violence. Release.",
          "score": 21.8
        }
      ]
    }
  },
  "combined_evidences": [],
  "combined_utterance": "I'm sorry, but I couldn't find any relevant information to confirm or deny the
director of the 2007 film Jin Ji-hee was in.",
  "feedback": [
    "- Relevant: The response is on-topic and directly addresses the user's question about the director of the
    2007 film Jin Ji-hee was in. 100/100",
    "- Temporally Correct: The response correctly uses the past tense to describe the search for the
    information. 100/100",
    "- Natural: The response sounds natural and maintains a polite tone. 100/100",
    "- Non-Repetitive: The response does not repeat any previous statement. 100/100"
  ],
  "feedback_scores": [
    100,
    100,
    100,
    100
  ],
  "refined_utterance": "I'm sorry, but I couldn't find any relevant information to confirm or deny the
director of the 2007 film Jin Ji-hee was in.",
  "user_utterance": "#Question#: Who directed the 2007 film Jin Ji-hee was in? #Answer#: The director's name
was Kim Pil-sung. Is this answer to the question a result of hallucination? Please indicate your judgment as
either \"Yes\" or \"No,\" accompanied by a step-by-step explanation.",
  "agent_utterance": "I'm sorry, but I couldn't find any relevant information to confirm or deny the director
of the 2007 film Jin Ji-hee was in.",
  "engine": "gpt-35-turbo",
  "generate_engine": "gpt-35-turbo",
  "draft_engine": "gpt-35-turbo",
  "pipeline": "early_combine",
  "wall_time_seconds": 20.4
}
```

Table 15: An illustrative example of the process in WikiChat for retrieving, generating, and refining responses, resulting in a final fact-checked judgment.

Table 16: Retrieval outcomes for specific and general queries for both correct and incorrect details, the relevant knowledge is highlighted in purple.

| Details for Verification | Top2 Retrieval Results for Specific Query | Top2 Retrieval Results for General Query |
|---|---|---|
| **Correct Detail**: Peace of Westphalia was signed in Mnster and Osnabrck, Germany. | **Query**: Was the Peace of Westphalia signed in Mnster and Osnabrck, Germany?

**Title**: Peace of Westphalia. **Article**: The Peace of Westphalia (, ) is the collective name for two peace treaties signed in October 1648 *in the Westphalian cities of Osnabrck and Mnster*. They ended the Thirty Years' War (16181648) and brought peace to the Holy Roman Empire, ...

**Title**: Peace of Westphalia. **Article**: In Mnster, negotiations took place between the Holy Roman Empire and France, as well as between the Dutch Republic and Spain who on 30 January 1648 signed a peace treaty ending the Eighty Years' War that was not part of the Peace of Westphalia ... | **Query**: Where was the Peace of Westphalia signed?

**Title**: Peace of Westphalia. **Article**: The Peace of Westphalia (, ) is the collective name for two peace treaties signed in October 1648 *in the Westphalian cities of Osnabrck and Mnster*. They ended the Thirty Years' War (16181648) and brought peace to the Holy Roman Empire, ... |
| **Wrong Detail**: Peace of Westphalia was signed in Peking, China. | **Query**: Was the Peace of Westphalia signed in Peking, China?

**Title**: Sino-Portuguese Treaty of Peking. **Article**: Portugal followed up on this agreement by sending an envoy to Beijing, where a treaty of amity and commerce based on the protocol was drawn up. On 1 December 1887, the Treaty of Peking was signed by Chinese representatives Yikuang (Prince Qing) ...

**Title**: Sino-Portuguese Treaty of Peking. **Article**: On 26 March 1887, Campbell and Portuguese Foreign Minister Henrique de Barros Gomes signed the four-point Lisbon Protocol: Art. 1st.A Treaty of friendship and commerce with the most favoured nation ... | **Title**: Breitenbach, Rhineland-Palatinate. **Article**: The Peace of Westphalia, a series of peace treaties signed between May and October 1648 in Osnabrck and Mnster, which brought the Thirty Years' War to an end, guaranteed the free practice of religion, and in the late 17th century, this law was also operative in Breitenbach, where it is likely that ... |
| **Correct Detail**: John Williams composed the score for "Star Wars." | **Query**: Did John Williams composed the score for "Star Wars"?

**Title**: Star Wars: The Rise of Skywalker (soundtrack). **Article**: On January 10, 2018, it was confirmed that John Williams would return to compose and conduct "The Rise of Skywalker". The next month, Williams announced that it would be the last "Star Wars" film for which he would compose the score. In August 2019, ...

**Title**: Star Wars (soundtrack). **Article**: Star Wars (Original Motion Picture Soundtrack) is the soundtrack album to *the 1977 film "Star Wars", composed and conducted by John Williams* and performed by the London Symphony Orchestra. Williams' score ... | **Query**: Who composed the score for "Star Wars"?

**Title**: Star Wars (soundtrack). **Article**: Star Wars (Original Motion Picture Soundtrack) is the soundtrack album to *the 1977 film "Star Wars", composed and conducted by John Williams* and performed by the London Symphony Orchestra. Williams' score ... |
| **Wrong Detail**: Christopher Nolan composed the score for "Star Wars." | **Query**: Did Christopher Nolan composed the score for "Star Wars"?

**Title**: List of awards and nominations received by Ludwig Gransson. **Article**: He received a second Academy Award nomination for Best Original Song thanks to "Lift Me Up", performed by Rihanna and written for the soundtrack of "Wakanda Forever". In 2020, he worked with Christopher Nolan in the film "Tenet", for which ...

**Title**: Tales of the Jedi (TV series). **Article**: Additional music for the series is composed by Sean Kiner, Deana Kiner, David Glen Russell, Nolan Markey and Peter Lam. Walt Disney Records released the soundtrack for the first season of "Tales of the Jedi" digitally on October 26, 2022, alongside the ... | **Title**: Music of Star Wars. **Article**:James L. Venable and Paul Dinletir composed the music of (20032005) 2D animated series, Ryan Shore serves as the composer for "Star Wars: Forces of Destiny" (20172018) and "Star Wars Galaxy of Adventures" (20182020), and Michael Tavera composes the score to ... |

Table 17: Performance comparison of different methods for hallucination detection in QA task, evaluated based on different version of GPT-3.5. Results of methods using external ground truth knowledge (i.e., knowledge provided by HaluEval) are shown inside the parentheses, and results generated based on Wiki knowledge are shown outside the parentheses.

| Model | Method | TPR (%) | TNR (%) | Avg Acc (%) |
|---|---|---|---|---|
| gpt-3.5-turbo-0613 | HaluEval (Vanilla) | 37.9 | 82.8 | 60.35 |
| | HaluEval (CoT) | 66.2 | 35.7 | 50.95 |
| | HaluEval (Knowledge) | 49.8 (42.6) | 65.3 (92.7) | 57.55 (67.65) |
| | KnowHalu (Structured) | 65.3 (70.4) | 56.9 (83.8) | 61.10 (77.10) |
| | KnowHalu (Unstructured) | 68.2 (67.5) | 46.7 (70.7) | 57.45 (69.10) |
| | KnowHalu (Aggregation) | **68.4 (71.7)** | **61.0 (85.7)** | **64.70 (78.70)** |
| gpt-3.5-turbo-1106 | HaluEval (Vanilla) | 44.1 | 55.2 | 48.15 |
| | HaluEval (CoT) | 66.5 | 21.6 | 44.05 |
| | HaluEval (Knowledge) | 34.4 (38.1) | 71.7 (75.7) | 53.05 (56.90) |
| | KnowHalu (Structured) | 72.6 (75.7) | 66.6 (80.0) | 69.60 (77.85) |
| | KnowHalu (Unstructured) | 77.3 (68.9) | 53.2 (75.7) | 65.25 (72.30) |
| | KnowHalu (Aggregation) | **76.3 (77.5)** | **67.8 (83.1)** | **72.05 (80.30)** |

Table 18: Performance comparison of KnowHalu using different query formulations, evaluated using the Starling-7B model and off-the-shelf knowledge for the QA task. The table reports the True Positive Rate (TPR), Abstain Rate for Positive cases (ARP), True Negative Rate (TNR), Abstain Rate for Negative cases (ARN), and Average Accuracy (Avg Acc) for each method. The optimal results for structured knowledge and unstructured knowledge are respectively highlighted in bold.

| Formulation | Method | TPR | ARP | TNR | ARN | Avg Acc (%) |
|---|---|---|---|---|---|---|
| Specific Query | KnowHalu (Structured) | 60.7 | 12.8 | **86.6** | **6.1** | 73.65 |
| | KnowHalu (Unstructured) | 70.3 | **3.5** | **86.5** | **4.5** | 78.40 |
| General Query | KnowHalu (Structured) | **67.8** | 8.6 | 83.1 | 8.3 | **75.45** |
| | KnowHalu (Unstructured) | **72.4** | 4.2 | 85.9 | 5.5 | **79.15** |
| Combined Queries | KnowHalu (Structured) | 65.8 | **7.9** | 84.5 | 6.9 | 75.15 |
| | KnowHalu (Unstructured) | 68.9 | 5.1 | 84.2 | 5.3 | 76.55 |

Table 19: Performance comparison of KnowHalu using different query formulations, evaluated using the Starling-7B model for the Text Summarization task, the $K$ is set to 3. The table reports the True Positive Rate (TPR), True Negative Rate (TNR), and Average Accuracy (Avg Acc) for each method. The optimal results for structured knowledge and unstructured knowledge are respectively highlighted in bold.

| Formulation | Method | TPR | TNR | Avg Acc (%) |
|---|---|---|---|---|
| Specific Query | KnowHalu (Structured) | 80.0 | 45.2 | 62.6 |
| | KnowHalu (Unstructured) | 67.6 | 61.6 | 64.6 |
| General Query | KnowHalu (Structured) | **87.8** | 34.2 | 61.0 |
| | KnowHalu (Unstructured) | **84.0** | 41.8 | 62.9 |
| Combined Queries | KnowHalu (Structured) | 80.2 | **45.4** | **62.8** |
| | KnowHalu (Unstructured) | 65.0 | **67.2** | **66.1** |

Table 20: Performance comparison of KnowHalu using different number of retrieved passages $K$, evaluated based on the Starling-7B model for Text Summarization Task. The table reports the True Positive Rate (TPR), True Negative Rate (TNR), and Average Accuracy (Avg Acc) for each method. The optimal results for structured knowledge and unstructured knowledge are respectively highlighted in bold.

| Top-$K$ Passages | Method | TPR | TNR | Avg Acc (%) |
|---|---|---|---|---|
| $K = 1$ | KnowHalu (Structured) | **84.8** | 39.8 | 62.3 |
| | KnowHalu (Unstructured) | **76.2** | 49.6 | 62.9 |
| $K = 2$ | KnowHalu (Structured) | 81.0 | 43.8 | 62.4 |
| | KnowHalu (Unstructured) | 68.0 | 62.0 | 65.0 |
| $K = 3$ | KnowHalu (Structured) | 80.2 | **45.4** | **62.8** |
| | KnowHalu (Unstructured) | 65.0 | 67.2 | **66.1** |
| $K = 4$ | KnowHalu (Structured) | 79.6 | 45.2 | 62.4 |
| | KnowHalu (Unstructured) | 62.0 | 68.6 | 65.3 |
| $K = 5$ | KnowHalu (Structured) | 81.0 | 43.8 | 62.4 |
| | KnowHalu (Unstructured) | 58.8 | **70.4** | 64.6 |

Table 21: Detailed selection of thresholds $\delta_1$ and $\delta_2$, along with the corresponding quantiles, determined on the validation set for the best setting of each task. The query formulation presented in the table is used for the base judgment. Typically, the supplementary judgment adopts the same formulation to achieve the optimial performance, unless otherwise specified.

| Task | Model | Knowledge Source | Formulation of Query | Knowledge Form for Base Judgment | $\delta_1$ ($q_1$) | $\delta_2$ ($q_2$) |
|------|-------|------------------|----------------------|----------------------------------|--------------------|--------------------|
| QA | Starling-7B | off-the-shelf knowledge | General Query | Unstructured Knowledge | 0.986236 (0.10) | 0.995475 (0.25) |
| | | Wiki retrieval knowledge | Combined Query | Unstructured Knowledge | 0.999455 (0.70) | 0.999241 (0.70) |
| | Mistral | off-the-shelf knowledge | General Query[1] | Unstructured Knowledge | 0.995717 (0.40) | 0.958954 (0.40) |
| | | Wiki retrieval knowledge | Combined Query | Unstructured Knowledge | 0.997418 (0.55) | 0.999343 (0.65) |
| | GPT-3.5 | off-the-shelf knowledge | Combined Query | Structured Knowledge | 0.999978 (0.70) | 0.999994 (0.75) |
| | | Wiki retrieval knowledge | Combined Query | Structured Knowledge | 0.999884 (0.30) | 0.999934 (0.30) |
| Text Summarization | Starling-7B | Original Document | Combined Query | Unstructured Knowledge | 0.999103 (0.25) | 0.999436 (0.30) |
| | GPT-3.5 | Original Document | Combined Query | Structured Knowledge | 0.999881 (0.20) | 0.999915 (0.45) |

[1] Here, as a special case, the supplementary judgment instead uses a combined query formulation.

