# OpenReview forum: "KnowHalu: Multi-Form Knowledge Enhanced Hallucination Detection"
_ICLR.cc/2025/Conference — Submitted to ICLR 2025_

### Official Review · Reviewer_NS6F · 2024-10-16

**Soundness:** 3
**Presentation:** 2
**Contribution:** 2
**Rating:** 6
**Confidence:** 3

**Summary:**

This paper proposes a KnowHalu method to detect hallucination, which is roughly divided into two stages, including off-target hallucination detection and multi-step facutal checking. The authors conducted experiments on multi-hop QA and text summariztion tasks on the HaluEval dataset. Three LLMs including Mistral-7B, Starling-7B and gpt-3.5-turbo-1106 were used. The experimental results show that the accuracy is improved compared with baseline models such as self-consistency that do not rely on retrieval evidence.

**Strengths:**

1. The authors conducted sufficient experiments and the method description is generally clear. The motivation of the proposed method is intuitive
2. The paper includes in-depth analyses of various components of KnowHalu. These studies may provide valuable insights into the contribution of each component.

**Weaknesses:**

1. Although the proposed multi-step reasoning process decomposes the query into multiple sub-queries to improve the detection accuracy, it involves multiple steps of knowledge retrieval, reasoning and aggregation, but it introduces engineering complexity and difficulty in reproduction, such as shared key-value caching and multi-form queries. I suggest that the authors open source the code and provide detailed reproducible instructions. While some latency comparisons are provided, a more thorough analysis of the computational overhead compared to simpler approaches would be valuable, especially for real-time applications.
2. The evaluation relies heavily on the HaluEval dataset, where hallucinations are generated by chatgpt and may not fully represent hallucination behavior in real-world scenarios. I would like the authors to test on a wider dataset with more sources of hallucinations or real sources of hallucinations (1000/500 given the limited size of the test set).
3. In addition, I hope that the author conduct manual evaluation tests on the detected hallucinations and reasoning processes, and conduct corresponding error analysis to gain insight into their limitations and potential improvement directions, such as errors in early stages (query or knowledge retrieval, rewriting queries) may be amplified through the pipeline. Robustness analysis of cascading errors is needed.
4. The authors are encouraged to take a systematic exploration of how different scales of model size affect performance.
5. The writing and presentation still need improvement. Such as Line 222 - one-hope, the size of Figure 7, Table 10, Table 15

**Questions:**

1. Is there a clear and strict definition of Off-Target Hallucination?
2. What is the reason for the performance difference of two query formulations? How are the two queries generated?

---

> ### Author Response · Authors · 2024-11-22
> **Response to Reviewer NS6F (Part-1)**
>
> We are deeply grateful to the reviewer for their insightful and thorough feedback, and we appreciate the recognition of our work's significance! The suggestions and comments made for our work have significantly helped to improve its quality.
>
> > **Q1: Open source the code and provide detailed reproducible instructions.**
>
> Thank you for recognizing our work! We guarantee that we will open source the code and provide detailed instructions for reproducing the results shown in the paper as clearly as possible once it is accepted.
>
> > **Q2: While some latency comparisons are provided, a more thorough analysis of the computational overhead compared to simpler approaches would be valuable, especially for real-time applications.**
>
> Thank you for your insightful suggestion and for highlighting this important aspect. We fully recognize the critical role that latency plays in real-time applications and understand the necessity of balancing computational efficiency with detection accuracy.
>
> Our approach is designed to prioritize robust detection, particularly in scenarios where minimizing hallucinations is paramount. In such cases, the slight increase in computational overhead is justified by the enhanced reliability our method provides. However, we also acknowledge that for applications requiring rapid initial detections with lower computational demands, simpler approaches may indeed be more suitable.
>
> To address your feedback, we will include a more detailed analysis of the computational overhead in our revision. Thank you again for your valuable comment, which will help us improve the applicability of our work.
>
> > **Q3: I would like the authors to test on a wider dataset with more sources of hallucinations or real sources of hallucinations (1000/500 given the limited size of the test set).**
>
> Thank you for your valuable suggestion. In our current work, for the QA task, we utilize 1,000 correct answers and 1,000 hallucinated answers. Similarly, for the summarization task, we have 500 accurate summaries and 500 hallucinated summaries. Our primary focus has been on knowledge-enhanced hallucination detection. To address your concern, we explored an additional popular WikiQA dataset [1] to achieve a broader evaluation. We selected questions with correct answers and complemented them with randomly sampled incorrect answers for the same questions within the dataset. This approach ensured that each question was paired with one correct and one incorrect answer, resulting in a final set of 857 questions with both correct and incorrect answers. For example, a sample QA pair from WikiQA is:
>
> **Q:** How are glacier caves formed?
> **A (correct):** A glacier cave is a cave formed within the ice of a glacier.
> **A (hallucinated):** Glacier caves are often called ice caves, but this term is properly used to describe bedrock caves that contain year-round ice.
>
> Here are the final results:
> **With the Starling-7B model:**
>
> | Method                  | TPR (%)   | TNR (%)   | Avg Acc (%) |
> | ----------------------- | --------- | --------- | ----------- |
> | HaluEval (CoT)          | 39.21     | 82.38     | 60.79       |
> | HaluEval (Knowledge)    | 23.80     | **76.55** | 50.18       |
> | KnowHalu (Structured)   | **87.16** | 58.59     | 72.93       |
> | KnowHalu (Unstructured) | 87.05     | 63.71     | 75.38       |
> | KnowHalu (Aggregation)  | 86.46     | 64.99     | **75.73**   |
>
> **With GPT-3.5:**
>
> | Method                  | TPR (%)   | TNR (%)   | Avg Acc (%) |
> | ----------------------- | --------- | --------- | ----------- |
> | HaluEval (CoT)          | 56.24     | 60.56     | 58.40       |
> | HaluEval (Knowledge)    | 22.64     | 54.61     | 38.62       |
> | KnowHalu (Structured)   | 88.68     | **64.64** | 76.66       |
> | KnowHalu (Unstructured) | **91.02** | 57.06     | 74.04       |
> | KnowHalu (Aggregation)  | 89.26     | **64.64** | **76.95**   |
>
> We directly adopted the aggregation hyperparameters used in the HaluEval dataset, which may lead to suboptimal performance for KnowHalu (Aggregation) due to distribution shifts. Nevertheless, as demonstrated with GPT-3.5, our method still provides an 18.55% improvement over the baselines.
>
> Thank you again for your insightful feedback. We believe these additional evaluations strengthen our work and address your concerns effectively.
>
> [1] Yang, Y., Yih, W. T., & Meek, C. (2015, September). Wikiqa: A challenge dataset for open-domain question answering. In Proceedings of the 2015 conference on empirical methods in natural language processing (pp. 2013-2018).

---

> > ### Author Response · Authors · 2024-11-22
> > **Response to Reviewer NS6F (Part-2)**
> >
> > > **Q4: In addition, I hope that the author conduct manual evaluation tests on the detected hallucinations and reasoning processes, and conduct corresponding error analysis to gain insight into their limitations and potential improvement directions, such as errors in early stages (query or knowledge retrieval, rewriting queries) may be amplified through the pipeline. Robustness analysis of cascading errors is needed.**
> >
> > Thanks for your valuable feedback! We have performed a manual evaluation of the reasoning processes involved in detecting hallucinations. We indeed observed some errors occurring in the early stages—such as during query formulation, knowledge retrieval, or query rewriting—can be amplified as they propagate through the pipeline sometimes.
> >
> > Currently, our approach can mitigate these errors to some extent, but it cannot entirely eliminate them. This limitation arises because certain steps are executed by LLMs, which inherently introduce some potential variability. Additionally, implementing more rigorous verification at each stage would increase processing latency, impacting overall efficiency.
> >
> > We acknowledge the importance of conducting a thorough robustness analysis of these cascading errors. In our revision, we will provide a detailed analysis highlighting the nature and impact of these errors, along with potential strategies for further improvement. Thank you once again for your insightful feedback. We are committed to addressing these concerns to strengthen our work.
> >
> > > **Q5: The authors are encouraged to take a systematic exploration of how different scales of model size affect performance.**
> >
> > Thank you for your insightful suggestion. In response, we have expanded our study to include two additional models: LLaMA 3.1-75B and LLaMA 3.1-405B. We use the QA task and the knowledge here comes from the Wiki database. Our findings indicate a linear scaling relationship between model size and performance metrics. Below are the results showcasing this trend:
> >
> > For Mistral-7B model,
> >
> > | Method                  | TPR (%)  | TNR (%)  | Inconclusive Rate(%) | Avg Acc (%) |
> > | ----------------------- | -------- | -------- | -------------------- | ----------- |
> > | HaluEval (Knowledge)    | 50.9     | 47.9     | -                    | 49.40       |
> > | KnowHalu (Structured)   | 54.0     | **67.3** | 12.35                | 60.65       |
> > | KnowHalu (Unstructured) | **63.5** | 58.4     | 11.70                | **60.95**   |
> >
> > For LLaMA3.1-70B model,
> >
> > | Method                  | TPR (%)  | TNR (%)  | Inconclusive Rate(%) | Avg Acc (%) |
> > | ----------------------- | -------- | -------- | -------------------- | ----------- |
> > | HaluEval (Knowledge)    | 86.0     | 45.3     | -                    | 65.65       |
> > | KnowHalu (Structured)   | **78.7** | 71.9     | 8.35                 | **75.30**   |
> > | KnowHalu (Unstructured) | 74.0     | **75.4** | 10.80                | 74.70       |
> >
> > For LLaMA3.1-405B model,
> >
> > | Method                  | TPR (%) | TNR (%)  | Inconclusive Rate(%) | Avg Acc (%) |
> > | ----------------------- | ------- | -------- | -------------------- | ----------- |
> > | HaluEval (Knowledge)    | 85.8    | 45.6     | -                    | 65.70       |
> > | KnowHalu (Structured)   | 74.2    | 74.5     | 11.25                | 74.35       |
> > | KnowHalu (Unstructured) | 73.7    | **76.9** | 8.75                 | **75.30**   |
> >
> > As we can see, with larger model scales, the performance of KnowHalu, utilizing wiki knowledge, also improves significantly. We will add this systematic exploration of how different model scales affect performance and include a corresponding discussion in our revision. Thank you for bringing this to our attention!
> >
> > > **Q6: The writing and presentation still need improvement. Such as Line 222 - one-hope, the size of Figure 7, Table 10, Table 15**
> >
> > Thank you for pointing out these issues. We apologize for the typo and we have resized Figure 7, Table 10, and Table 15 to improve their clarity and presentation. We appreciate your careful reading and constructive feedback!

---

> > > ### Author Response · Authors · 2024-11-22
> > > **Response to Reviewer NS6F (Part-3)**
> > >
> > > > **Q7: Is there a clear and strict definition of Off-Target Hallucination?**
> > >
> > > Apologies for any confusion. As outlined in Table 1, we categorize off-target hallucinations into five different types, ranging from “Vague or Broad Answers” to “Overgeneralization or Simplification.” In essence, an off-target hallucination occurs when a response is factually correct but does not appropriately address the specific question asked. For example, if the question is “What is the capital of France?” and the response is “France is in Europe,” the statement itself is accurate, but it fails to answer the question and is actually irrelevant (“off-target”) with the question, thereby constituting an off-target hallucination.
> > >
> > > To provide a clearer and more precise definition, we can consider off-target hallucinations as those types of hallucinations that exclude factual hallucinations. In other words, off-target hallucinations involve responses that may be factually correct but still do not appropriately address the question posed. This definition serves as the complement to factual hallucination, focusing on the relevance and appropriateness of the response rather than its factual accuracy.
> > >
> > > We hope this clarification helps in understanding the distinction and definition of off-target hallucinations in our work.
> > >
> > > > **Q8: What is the reason for the performance difference of two query formulations? How are the two queries generated?**
> > >
> > > Thank you for your valuable question! We explore the retrieval performance differences between two query formulations in Table 16. The key difference lies in the details contained within the queries. For example, a specific query like “Did *John Williams* compose the score for ‘Star Wars’?” allows for accurate retrieval from the database because it contains semantic details that match available information. In contrast, if the query is based on incorrect details from the wrong answer, such as “Did *Christopher Nolan* compose the score for ‘Star Wars’?”, it fails to retrieve relevant passages since the database lacks such information. In this scenario, a more general query like “Who composed the score for ‘Star Wars’?” yields more accurate retrieval because it does not assume specific, potentially incorrect, details.
> > > Therefore, the performance variation between the two query types depends on the accuracy of the details provided in the question. A specific query can precisely locate information to verify a correct answer, but it leads to poor retrieval performance if the details in the answer are incorrect. A general query, which does not depend on the specifics of the answer, often provides better results in such cases. In our work, we combine both types of queries to optimize retrieval performance.
> > > The generation of these two queries is handled by the LLM during the Step-wise Reasoning and Query stage, as illustrated in Figure 1. The LLM is prompted (few-shot) to generate both a specific and a general query at the same time, for example, "#Query-1#: Was 'Star Wars' the 1977 space-themed movie in which the character Luke Skywalker first appeared? [Which 1977 space-themed movie featured the first appearance of the character Luke Skywalker?]," where the first part is the specific query and the bracketed text is the general query.

---

### Official Review · Reviewer_H9RQ · 2024-10-31

**Soundness:** 3
**Presentation:** 2
**Contribution:** 2
**Rating:** 5
**Confidence:** 4

**Summary:**

This paper presents a novel approach, referred to as KnowHallu, for detecting hallucinations in texts generated by large language models (LLMs). Firstly, it isolates a new type of hallucination, referred to as the off-target hallucination, and then it conducts a novel multi-form knowledge-based fact-checking through a comprehensive four-stage pipeline. Experimental results demonstrate the effectiveness of the proposed framework for detecting hallucinations.

**Strengths:**

1.The figure is presented in a clear and organized manner, effectively incorporating a wealth of details that contribute to KnowHallu’s comprehensibility.

2.The proposed framework demonstrates effectiveness based on a substantial number of experiments conducted.

**Weaknesses:**

1.Lack of clarification of the concept of ‘off-target’ hallucinations. The author claims in the abstract that they introduce a new category of hallucinations. However, it is noteworthy that prior research, such as HaluEval, has already categorized model responses into "unverifiable", "non-factual" and "irrelevant" classifications. Furthermore, earlier studies focusing on fact-checking have highlighted that not all statements are suitable for this process, which has led to the emergence of the claim detection task. Therefore, why is the concept of off-target hallucination emphasized in this context? Additionally, I am interested to know whether there is statistical data in Table 1 regarding the percentage of different types of questions present in actual LLM responses.

2.Usage of testing tasks. The original HaluEval framework includes both Dialogue and General scenarios; however, it is unclear why these two scenarios were not included in the current evaluation. The proposed design of step-wise reasoning is a common approach in addressing multi-hop questions. However, it raises the question of its applicability across all hallucination detection tasks.

**Questions:**

See weakness.

1.Regarding the experimental results presented in Table 2, it is noteworthy that GPT-3.5 does not exhibit superior performance compared to other models. Isn't that somewhat counterintuitive?

2.The overall framework appears to combine various manually designed prompting strategies with established fact-checking techniques. For example, the approach involves decomposing the question into smaller components and verifying each part through retrieval on a one-by-one basis. From your view, what's the difference in the current hallucination detection task and the original fact-checking task?

I would like to improve my score if my aforementioned questions are answered properly.

---

> ### Author Response · Authors · 2024-11-22
> **Response to Reviewer H9RQ (Part-1)**
>
> We are deeply grateful to the reviewer for their thorough and insightful feedback. Their expertise and dedicated time have significantly contributed to improving the quality of this work!
>
> > **Q1:  The author claims in the abstract that they introduce a new category of hallucinations. However, it is noteworthy that prior research, such as HaluEval, has already categorized model responses into "unverifiable", "non-factual" and "irrelevant" classifications.**
>
> Thank you for your insightful comment. We apologize for any confusion regarding the categorization of hallucinations in prior work such as HaluEval.
>
> In HaluEval, the authors indeed utilize "non-factual" and "irrelevant" categories to generate hallucinated data. However, during the evaluation, tasks—such as the [QA](https://github.com/RUCAIBox/HaluEval/blob/main/evaluation/qa/qa_evaluation_instruction.txt)  task—they still simply employs a binary classification setting, i.e., just only determining whether an answer is a hallucination or not, and there is also no class for "unverifiable" category (while we have one category called “INCONCLUSIVE” for it).
>
> In contrast, our work introduces a more nuanced categorization of hallucinations by refining these existing classes and incorporating additional categories. This finer-grained classification enables more accurate detection of different types of hallucinations. Specifically, our approach achieves a 15-20% improvement in detection accuracy over the original HaluEval framework. Furthermore, in scenarios where recall for hallucination is more important, we could just simply map all “unverifiable” cases to hallucinations, which would be aligned with the same binary classification setting, the improvement would then be 20-25% across all models over the original HaluEval on QA task.
>
> To clarify, our abstract refers to the first introduction of these new categories to enhance hallucination detection precision. While we acknowledge that prior research like HaluEval has already identified broad categories, our contribution lies in making this categorization more detailed and developing specialized detection strategies for them. We appreciate the reviewer’s recognition of the importance of finer categorization for improved hallucination detection and will ensure that this distinction is more clearly articulated in the revised abstract. Thank you again for your valuable feedback!
>
> > **Q2:  Additionally, I am interested to know whether there is statistical data in Table 1 regarding the percentage of different types of questions present in actual LLM responses.**
>
> Thank you for the insightful observation! Here is the statistical data for hallucinated data in QA:
>
> | Vague or Broad Answers | Parroting or Reiteration | Misinterpretation of Question | Negation or Incomplete Information | Overgeneralization or Simplification | Fabrication |
> | ---------------------- | ------------------------ | ----------------------------- | ---------------------------------- | ------------------------------------ | ----------- |
> | 6.6%                   | 3.1%                     | 11.1%                         | 11.6%                              | 0.6%                                 | 67.0%       |
>
> As we can see, aside from pure fabrication (which can be detected by fact-checking), there are still a lot of other types of hallucinations, i.e., off-target hallucination cases.

---

> > ### Author Response · Authors · 2024-11-22
> > **Response to Reviewer H9RQ (Part-2)**
> >
> > > **Q3: Furthermore, earlier studies focusing on fact-checking have highlighted that not all statements are suitable for this process, which has led to the emergence of the claim detection task. Therefore, why is the concept of off-target hallucination emphasized in this context?**
> >
> > Thank you for the valuable feedback! Traditional claim detection methods primarily focus on identifying factual inaccuracies or fabrications within responses. However, this focus overlooks scenarios where the response is factually correct but irrelevant or unaligned with the original query—i.e., the off-target hallucination.
> >
> > For example, consider the query: "What is the capital of France?" A response such as "France is in Europe" is indeed factually accurate as an individual claim, thus traditional claim detection methods would naturally recognize this statement as correct and not label it as a hallucination. However, this response fails to address the actual question, making it an off-target hallucination. In other words, when evaluated in isolation, the claim itself appears accurate, but in the context of the query, it is an obvious off-target hallucination.
> >
> > Specifically, in our comparison with the baseline WikiChat, which employs claim-based verification, we also indeed observed that WikiChat correctly verifies factual claims extracted from responses. However, it fails to identify off-target hallucinations when the response, although factually correct as individual claims, does not solve the original question. As a result, WikiChat performs somewhat poorly in HaluEval, and we also provide a case study on Page 27 illustrating why pure claim detection is insufficient for hallucination detection when using WikiChat.
> >
> > Besides, from the perspective of pure empirical performance, as presented in Table 6, it demonstrates that incorporating off-target hallucination detection (Stage-1) enhances overall accuracy by 7~8% compared to approaches relying solely on direct fabrication checking (Stage-2). This improvement underscores the significance of identifying and addressing off-target hallucinations to achieve more reliable fact-checking outcomes.
> >
> > In short, there are many existing studies [1,2,3,4] emphasize that not all hallucinations stem from factual inaccuracies; some arise from a lack of relevance or directness in addressing the query. Therefore, by distinguishing off-target hallucinations from straightforward fabrications, we can achieve a more comprehensive and accurate hallucination detection of responses, and that’s why we emphasize it in our paper.
> >
> > [1] Huang, L., Yu, W., Ma, W., Zhong, W., Feng, Z., Wang, H., ... & Liu, T. (2023). A survey on hallucination in large language models: Principles, taxonomy, challenges, and open questions. arXiv preprint arXiv:2311.05232.
> >
> > [2] Zhang, Y., Li, Y., Cui, L., Cai, D., Liu, L., Fu, T., ... & Shi, S. (2023). Siren's song in the AI ocean: a survey on hallucination in large language models. arXiv preprint arXiv:2309.01219.
> >
> > [3] Li, J., Cheng, X., Zhao, X., Nie, J. Y., & Wen, J. R. (2023, December). Halueval: A large-scale hallucination evaluation benchmark for large language models. In The 2023 Conference on Empirical Methods in Natural Language Processing.
> >
> > [4] Zheng, S., Huang, J., & Chang, K. C. C. (2023). Why does chatgpt fall short in providing truthful answers. ArXiv preprint, abs/2304.10513.

---

> > > ### Author Response · Authors · 2024-11-22
> > > **Response to Reviewer H9RQ (Part-3)**
> > >
> > > > **Q4: The original HaluEval framework includes both Dialogue and General scenarios; however, it is unclear why these two scenarios were not included in the current evaluation. The proposed design of step-wise reasoning is a common approach in addressing multi-hop questions. However, it raises the question of its applicability across all hallucination detection tasks.**
> > >
> > > We apologize for any confusion regarding the inclusion of Dialogue and General scenarios from the original HaluEval framework in our current evaluation. Our primary focus in this work is on knowledge-enhanced hallucination detection. In the general scenarios of HaluEval, the questions are like "Design a shape with 10 vertices (corners)," and "Generate a web page for a book review website." For these types of questions, the detection of hallucinations does not rely on external knowledge sources like Wikipedia. These questions are inherently different from knowledge-intensive tasks and fall outside the scope of the hallucination cases we aim to address in our paper. Therefore, we do not include the general scenarios.
> > >
> > > Regarding the [dialogue](https://github.com/RUCAIBox/HaluEval/blob/main/data/dialogue_data.json) scenarios, we initially attempted to apply our method to this task. However, during our analysis, we identified significant issues with the quality of ground labels within the dialogue datasets. Specifically, our manual verification revealed that  around 20%~30% of dialogue history entries labeled as correct actually contained factual errors (especially for those errors coming from the intermedia response from the assistant in the dialogue, but they do not label them as hallucinations). Similarly, a notable portion of entries labeled as hallucinated did not exhibit factual errors upon manual checking. Besides, the knowledge provided for dialogue is also just a simple sentence, which is also not enough for verifying the correctness of the whole long dialogue history. These labeling inconsistencies compromise the reliability of the evaluation and, consequently, we have decided not to include it in the paper.
> > >
> > > But for reference, we still put the corresponding experimental result in dialogue as below:
> > >
> > > | Method               | TPR(%) | TNR(%) | Avg Inconclusive Rate(%) | Avg Acc(%) |
> > > | -------------------- | ------ | ------ | ------------------------ | ---------- |
> > > | HaluEval (Cot)       | 93.4   | 31.3   | -                        | 62.35      |
> > > | HaluEval (Knowledge) | 92.6   | 33.8   | -                        | 63.20      |
> > > | Self Consistency     | 89.4   | 39.5   | -                        | 64.45      |
> > > | SelfCheckGPT         | 24.5   | 75.3   | -                        | 49.90      |
> > > | KnowHalu             | 82.4   | 50.7   | 7.3                      | **66.55**  |
> > >
> > > For applicability, we are committed to enhancing the adaptability of our method across various hallucination detection tasks. To facilitate this, we will provide clear adaptation instructions once our code is released. Additionally, we will also include the detection for dialogue tasks in the code.

---

> ### Author Response · Authors · 2024-11-22
> **Response to Reviewer H9RQ (Part-4)**
>
> > **Q5: Regarding the experimental results presented in Table 2, it is noteworthy that GPT-3.5 does not exhibit superior performance compared to other models. Isn't that somewhat counterintuitive?**
>
> Thank you for your insightful question and for pointing out the observed performance of GPT-3.5 in Table 2. We apologize for any confusion caused. There are two possible reasons why GPT-3.5 does not exhibit superior performance compared to other models like Starling-7B:
>
> 1. **Competitive Performance:** First, GPT-3.5 does not necessarily have to outperform all other models. As shown on the [Arena Leaderboard](https://huggingface.co/spaces/lmarena-ai/chatbot-arena-leaderboard), the arena score of GPT-3.5-Turbo-1106 is 1068, which is actually even lower than the arena score of the model Starling-LM-7B-alpha, which is 1088. This is consistent with the performance comparisons presented in our paper. Besides, in Appendix C, we also show that newer versions of GPT-3.5 also exhibit a decline in performance, which is also consistent with [1].
>
> 2. **Handling of Requests and Query Behavior:** As shown in Appendix D of our paper, GPT-3.5 occasionally rejects certain requests, especially those identified as containing inappropriate or potentially invasive content. While this behavior was not observed for other models, leading to a degradation in its overall performance. Additionally, GPT-3.5 tends to exhibit a more `lazy' approach during the step-wise querying process, which can further impact its effectiveness compared to other models. As a result, the average inconclusive rate of GPT-3.5 is approximately 5% higher than other models.
>
> We appreciate your question and will incorporate additional clarifications in our paper to better explain why GPT-3.5 does not outperform other models. Your feedback helps us improve the clarity and comprehensiveness of our work.
>
> [1] Chen, L., Zaharia, M., & Zou, J. (2023). How is ChatGPT's behavior changing over time?. arXiv preprint arXiv:2307.09009.
>
> > **Q6: The overall framework appears to combine various manually designed prompting strategies with established fact-checking techniques. For example, the approach involves decomposing the question into smaller components and verifying each part through retrieval on a one-by-one basis. From your view, what's the difference between the current hallucination detection task and the original fact-checking task?**
>
> Thanks for the insightful question! We will articulate more clearly the motivations behind our methodology and the specific contributions, as outlined below:
>
> 1. **Two-Phase Hallucination Detection:**
>
> We first introduce a two-phase detection strategy for both off-target and fabrication hallucinations. This distinction is crucial because off-target types, such as broad answers, are considered hallucinations despite being factually correct, and thus cannot be detected through standard fabrication detection methods. A separate detection stage is thus essential for these hallucinations before proceeding to factual checking. We will further expand on this motivation in Section 3.1 of our revision.
>
> 2. **Off-target Hallucination:**
>
> Our framework first categorizes specific off-target hallucinations and treats their detection as an extraction task, significantly improving detection accuracy. For instance, incorporating this specialized detection led to an approximate 7~8% increase in accuracy across all three models tested, as demonstrated in Table 6. This underscores the importance of identifying and accurately addressing these subtle but critical types of hallucinations.
>
> 3. **Fact-Checking Enhancements:**
>    - **Diverse Knowledge Forms:** We show that the form of knowledge (structured vs. unstructured) significantly influences the detection accuracy. For example, the Starling-7B model benefits more from non-structured knowledge, with a 3.70% improvement, while GPT-3.5 gains a 5.55% increase with structured knowledge. This underlines the importance of matching knowledge forms with a model’s inherent reasoning capabilities for enhanced fact-checking precision.
>
>    - **Query Formulation in Knowledge Retrieval:** The formulation of queries (specific vs. general) during the knowledge retrieval stage has a significant impact on performance. This strategic approach to query formulation, aimed at fetching relevant and accurate knowledge for fact-checking, can lead to a further 3.50% improvement in results.
>
> Together, these methodological innovations and strategic enhancements result in a notable 15~20% improvement in QA task performance. Our forthcoming revision will further elucidate these points, ensuring a coherent understanding of the motivations and insights our paper offers. We will make these discussions and contributions more clear in our revision!

---

> > ### Comment · Reviewer_H9RQ · 2024-11-25
> >
> > Thanks for the response, and I have improved the score.

---

> > > ### Author Response · Authors · 2024-11-25
> > > **Thank you for the insightful review!**
> > >
> > > Thank you for your positive feedback, we are glad to have addressed most of your issues. If you have any further questions or concerns about the experiments, please feel free to let us know. We greatly appreciate your time and effort in helping to polish our work!

---

> ### Author Response · Authors · 2024-11-30
> **Rebuttal Follow-Up**
>
> Dear Reviewer,
>
> We want to follow up on our discussion here.
> In our work, we aim to provide a unified hallucination detection framework based on two stages, five steps including various forms of knowledge and query formulations. We also offer a clearer motivation and definition for 'off-target' hallucinations, including a mathematical formulation following your suggestion and Reviewer NS6F, which has been acknowledged by other reviewers. In short, an off-target hallucination is a type of response generated by a language model where the content may be factually correct but fails to directly address the specific question or prompt.
>
> We have also carefully improved our work following your feedback and will extend our efforts to include more tasks, such as the dialogue task you proposed. Your suggestions and comments would be invaluable to us. Please let us know if you have further questions or suggestions, and we would be eager to discuss and further improve our work to help build a safer AI community.
>
> If our revisions meet your expectations, might you consider raising the score? We greatly appreciate your effort and expertise in helping to polish our work!

---

### Official Review · Reviewer_wyLP · 2024-11-04

**Soundness:** 2
**Presentation:** 3
**Contribution:** 3
**Rating:** 6
**Confidence:** 4

**Summary:**

This paper presents KnowHalu, a novel hallucination detection framework leveraging multi-form knowledge to enhance the factual accuracy of Large Language Models (LLMs) and mitigate the generation of hallucinated content. KnowHalu employs a two-phase approach: initially identifying off-target hallucinations through semantic alignment, followed by a multi-step, knowledge-based fact-checking process that includes reasoning, query decomposition, knowledge retrieval, optimization, judgment generation, and aggregation. Extensive evaluations demonstrate KnowHalu's superiority over state-of-the-art baselines, with over 15% improvement in question answering tasks. The study introduces the concept of off-target hallucinations and explores the impact of query formulations and knowledge forms on detection accuracy, proposing an aggregation method to refine judgments based on different knowledge forms. While KnowHalu shows promising results on standard datasets, the experimental section could be enhanced, particularly in handling more complex dialogues and longer responses, and the analysis could benefit from deeper insights into the nuances of hallucination detection.

**Strengths:**

The "Off-Target Hallucination Checking" concept is indeed compelling as it tackles a critical shortcoming in the output of language models. The detection pipeline for addressing these hallucinations is not only effective but also yields encouraging results across question answering and text summarization tasks. Furthermore, the methodology section is articulated in a clear and comprehensible manner, which enhances the understandability of the approach taken.

**Weaknesses:**

The shortcomings of this paper are quite apparent. First, it lacks a thorough analysis of the RAG data sources. For instance, it does not address the specific impact of the knowledge base (KB) data source on the results, nor does it consider how data missingness or redundancy might affect the outcomes. Additionally, it would be beneficial to explore whether data conflicts could impact the model's robustness.

Moreover, Section 3.2 presents a rather lengthy description of the methods. It would be advisable to streamline this section to enhance overall clarity.

**Questions:**

In Tables 4 and 5, does KnowHalu perform better in unstructured scenarios than in structured ones? However, doesn't Table 3 present the opposite conclusion?

---

> ### Author Response · Authors · 2024-11-22
> **Response to Reviewer wyLP (Part-1)**
>
> Many thanks to the reviewer for the thoughtful and detailed feedback. The expertise and time invested in this work have been instrumental in enhancing its quality.
>
> > **Q1: The analysis lacks a thorough examination of the RAG data sources. For instance, it does not address the specific impact of the knowledge base (KB) as a data source on the results.**
>
> Thank you for this insightful question! In our study, the selection of RAG data sources is carefully tailored to each specific task to ensure optimal performance. For the QA task, we utilize the Wikipedia database as our knowledge base because the questions in the dataset (i.e., HotpotQA[1]) are derived from Wikipedia articles. This alignment ensures that the KB is both relevant and comprehensive for effectively addressing the QA task. Similarly, for the summarization task, the knowledge base consists exclusively of the original documents that need to be summarized, providing direct and pertinent information necessary for generating accurate summaries. In our experiments, we consistently use the same knowledge base for both our proposed method and all baseline models.
>
> Regarding your suggestion to explore the impact of varying knowledge bases, we fully acknowledge its value and consider it an important direction for future research. However, for QA tasks, [Wikipedia Corpus](https://dumps.wikimedia.org/) currently stands as the only large-scale, high-quality open-source dataset available in academia. Utilizing Wikipedia dumps has also been a widely accepted standard in the RAG domain for QA tasks and is employed by leading methods and benchmarks such as WikiChat[2], FlagEmbedding[3], DPR[4], and KILT[5], etc.
>
> So, while exploring different knowledge bases could provide additional insights, relying on Wikipedia as our sole knowledge source aligns with current best practices and ensures the reliability and comparability of our results. We appreciate your recommendation and will consider it for future studies to further enhance our analysis!
>
> [1] Yang, Z., Qi, P., Zhang, S., Bengio, Y., Cohen, W., Salakhutdinov, R., & Manning, C. D. (2018). HotpotQA: A Dataset for Diverse, Explainable Multi-hop Question Answering. In Proceedings of the 2018 Conference on Empirical Methods in Natural Language Processing (pp. 2369-2380).
>
> [2] Semnani, S., Yao, V., Zhang, H., & Lam, M. (2023, December). WikiChat: Stopping the Hallucination of Large Language Model Chatbots by Few-Shot Grounding on Wikipedia. In Findings of the Association for Computational Linguistics: EMNLP 2023 (pp. 2387-2413).
>
> [3] Zhang, P., Xiao, S., Liu, Z., Dou, Z., & Nie, J. Y. (2023). Retrieve anything to augment large language models. arXiv preprint arXiv:2310.07554.
>
> [4] Karpukhin, V., Oguz, B., Min, S., Lewis, P., Wu, L., Edunov, S., ... & Yih, W. T. (2020, November). Dense Passage Retrieval for Open-Domain Question Answering. In Proceedings of the 2020 Conference on Empirical Methods in Natural Language Processing (EMNLP) (pp. 6769-6781).
>
> [5] Petroni, F., Piktus, A., Fan, A., Lewis, P., Yazdani, M., De Cao, N., ... & Riedel, S. (2021, June). KILT: a Benchmark for Knowledge Intensive Language Tasks. In Proceedings of the 2021 Conference of the North American Chapter of the Association for Computational Linguistics: Human Language Technologies (pp. 2523-2544).
>
> > **Q2: Section 3.2 provides a rather lengthy description of the methods. Streamlining this section would help improve overall clarity.**
>
> Thank you for your valuable feedback! We aimed to provide a detailed description of our methods in Section 3.2 to ensure clarity for readers who may be less familiar with the area. We understand that this level of detail might be lengthy for people quite familiar with the subject. In our revision, we will streamline this section to better balance detail and clarity. Thanks again for your helpful suggestion!

---

> ### Author Response · Authors · 2024-11-22
> **Response to Reviewer wyLP (Part-2)**
>
> > **Q3: Furthermore, it does not consider how data missingness or redundancy might affect the outcomes. It would also be valuable to explore whether data conflicts could influence the model's robustness.**
>
> Thank you for your insightful suggestion! We apologize for any confusion caused. Indeed, we have carefully considered the impact of data missingness and data conflicts in our analysis, as detailed on line 304 of the paper. Unlike the baseline models, which do not account for these scenarios, our approach introduces an additional judgment category called “INCONCLUSIVE.” This category represents the situations where the retrieved knowledge is insufficient to determine the correctness of a response.
>
> To illustrate, here are the inconclusive rates for both the Starling-7B and GPT-3.5 models in the QA task:
>
> **Inconclusive Rates with Starling-7B Model:**
>
> | Method             | Knowledge Source | Inconclusive rate for hallucinated cases | Inconclusive rate for correct cases | Average Inconclusive rate |
> | ------------------ | ---------------- | ---------------------------------------- | ----------------------------------- | ------------------------- |
> | KnowHalu (Structured)  | Off-the-shelf    | 7.9%                                     | 6.9%                                | 7.40%                     |
> | KnowHalu (Unstructured) | Off-the-shelf    | 5.1%                                     | 5.3%                                | 5.20%                     |
> | KnowHalu (Structured)  | Wiki             | 9.0%                                     | 12.0%                               | 10.50%                    |
> | KnowHalu (Unstructured) | Wiki             | 5.3%                                     | 8.2%                                | 6.75%                     |
>
> **Inconclusive Rates with GPT-3.5 Model:**
>
> | Method             | Knowledge Source | Inconclusive rate for hallucinated cases | Inconclusive rate for correct cases | Average Inconclusive rate |
> | ------------------ | ---------------- | ---------------------------------------- | ----------------------------------- | ------------------------- |
> | KnowHalu (Structured)  | Off-the-shelf    | 7.2%                                     | 7.0%                                | 7.10%                     |
> | KnowHalu (Unstructured) | Off-the-shelf    | 16.4%                                    | 12.6%                               | 14.5%                     |
> | KnowHalu (Structured)  | Wiki             | 6.2%                                     | 7.0%                                | 6.60%                     |
> | KnowHalu (Unstructured) | Wiki             | 11.0%                                    | 12.8%                               | 11.9%                     |
>
> As a result, the incorporation of inconclusive examples might initially seem to undermine the performance of our methods compared to baselines that only support binary classification (hallucinated or not). Since in scenarios where recall for hallucination is more important, we could just simply map all INCONCLUSIVE cases to hallucinations. This alignment with the same binary classification setting as the baselines could directly increase the average accuracy by an additional 5~10%.
>
> We will ensure that this is more clearly explained in our revision and will include the detailed statistics for the INCONCLUSIVE category. Thank you for bringing this to our attention!
>
> > **Q4: In Tables 4 and 5, does KnowHalu perform better in unstructured scenarios than in structured ones? However, doesn’t Table 3 suggest the opposite conclusion?**
>
> We apologize for any confusion regarding the performance differences observed in our tables. The primary factor is the model used: GPT-3.5 demonstrates superior performance in structured scenarios, while the open-source Starling-7b model excels in unstructured scenarios. Specifically, Tables 4 and 5 showcase KnowHalu's performance with Starling-7b, highlighting better results in unstructured contexts. Similarly, Table 3 presents results indicating that the Starling-7b model outperforms in unstructured scenarios, whereas the GPT model exhibits stronger performance in structured scenarios. Therefore, there is no contradiction here; the key difference lies in the models used. We will make this observation clearer in our revision and thank you for your valuable feedback!

---

> > ### Comment · Reviewer_wyLP · 2024-11-25
> > **Thanks for response!**
> >
> > Thanks very much for addressing my issue! I have raised my scores!

---

> > > ### Author Response · Authors · 2024-11-25
> > > **Thank you for the helpful review!**
> > >
> > > Thank you for your positive feedback and for updating your scores, and we greatly appreciate the time and effort you have dedicated to helping polishing our work!

---

### Author Response · Authors · 2024-11-22
**Revision Summarization**

We thank all the reviewers for their valuable feedback on our paper. We are pleased to note that all reviewers found our paper well-written and recognized it as providing a novel and systematic LLM hallucination detection framework, along with a significant performance improvement over other methods. We have made the following revisions following the reviewers' suggestions to further improve our work.

- We have added the detailed inconclusive rate, which is proposed for considering data missingness, following the suggestion of **Reviewer wyLP** in Table 2.
- We added more clarification of the motivation for introducing a new category of hallucination, i.e., off-target hallucination, which we believe helps build a comprehensive hallucination categorization for the community, following **Reviewer H9RQ** and **Reviewer NS6F**.
- We included an analysis of the percentage of different types of questions shown in Table 1, following **Reviewer H9RQ**.
- We added additional experimental results on dialogue data from HaluEval and WikiQA, following the suggestions from **Reviewer H9RQ** and **Reviewer NS6F**. We show that our method still outperforms all other baselines with these two additional datasets.
- We conducted a systematic exploration of how different scales of model size affect performance, following **Reviewer NS6F**. The results show that the performance of the knowledge increases with the scaling of the model size linearly.

We sincerely appreciate the reviewers for their precious time in reviewing our paper and look forward to any further discussion to improve the quality of our work!

---

### Meta-Review · Area_Chair_vzyi · 2024-12-22

**Metareview:**

This paper introduces KnowHalu, a novel framework for hallucination detection in outputs of large language models (LLMs). The method employs a two-phase approach: isolating off-target hallucinations through semantic alignment and performing multi-step, knowledge-based factual checking. KnowHalu introduces the concept of off-target hallucinations and demonstrates significant improvements over state-of-the-art baselines in both question answering and summarization tasks.

(a) Scientific Claims and Findings
KnowHalu is a multi-form knowledge-enhanced hallucination detection framework comprising two phases: detection of off-target hallucinations and multi-step factual checking. The authors propose a new category of hallucination, off-target hallucinations, characterized by responses that are factually correct but irrelevant or nonspecific to the query. Extensive experiments on question answering and summarization tasks demonstrate that KnowHalu outperforms state-of-the-art methods with improvements of over 15% in question answering and 6% in summarization accuracy.

(b) Strengths

- Novel categorization of hallucinations with the introduction of off-target hallucinations.
- Rigorous two-phase detection framework addressing both factual inaccuracies and relevance issues.
- Comprehensive experimental validation across diverse tasks, datasets, and models.

(c) Weaknesses

- Heavy reliance on HaluEval dataset, limiting generalizability to real-world scenarios.
- Computational complexity and cascading errors in the multi-step pipeline were identified but not fully resolved.
- Presentation issues and lack of clarity in some sections, although improved after revisions.
- Limited robustness analysis of cascading errors and real-time applicability.

(d) Decision: weak reject

While the paper proposes an interesting approach and demonstrates promising results and reviewers slightly improved their scores after rebuttal, critical issues remain that limit its readiness for acceptance. The reliance on a single dataset raises concerns about generalizability, and the computational complexity of the method challenges its practical applicability. Addressing these concerns with broader real-world evaluations and streamlined implementations could significantly strengthen the work. I encourage the authors to refine these aspects and resubmit in the future, as this line of research has substantial potential.

**Additional Comments On Reviewer Discussion:**

The discussion during the rebuttal period focused on several key concerns raised by the reviewers, which are summarized as follows:

(a) Dataset Reliance and Generalizability (Reviewer NS6F, H9RQ)

- Concern: Both reviewers noted that the heavy reliance on the HaluEval dataset limits the generalizability of the findings, as the dataset does not fully capture real-world hallucination scenarios. NS6F specifically suggested testing on a broader dataset or with real-world sources of hallucinations.
- Author Clarification: The authors expanded the evaluation to include WikiQA and reported additional results that demonstrated improvements over baselines. They acknowledged the need for further evaluation on more diverse datasets.
- Outcome: Reviewer NS6F appreciated the additional evaluations but maintained concerns about broader generalizability and kept their score unchanged. Reviewer H9RQ increased their score after seeing the new analyses.

(b) Computational Complexity and Practicality (Reviewer NS6F)

- Concern: NS6F pointed out the engineering complexity of the multi-step pipeline, including potential cascading errors and computational overhead. They requested a more detailed robustness and efficiency analysis, especially for real-time applications.
- Author Clarification: The authors provided latency comparisons and clarified that their approach could be optimized for deployment. They also detailed how cascading errors were mitigated but acknowledged that quantitative robustness analysis was limited.
- Outcome: Reviewer NS6F appreciated the clarifications but maintained concerns about practicality and left their score unchanged.

(c) Definition and Novelty of Off-Target Hallucinations (Reviewer H9RQ, NS6F)

- Concern: H9RQ questioned the novelty of the off-target hallucination concept, pointing to prior work like HaluEval, which categorized similar phenomena. NS6F asked for a more formal definition of off-target hallucinations.
- Author Clarification: The authors clarified their contribution by distinguishing their nuanced categorization and detection framework. They also provided a formal mathematical definition during the rebuttal.
- Outcome: Reviewer H9RQ appreciated the clarifications and increased their score. NS6F, while acknowledging the improvement, did not revise their score due to remaining concerns.

(d) Robustness and Error Analysis (Reviewer NS6F)

- Concern: NS6F emphasized the need for a detailed robustness analysis of cascading errors in the multi-step pipeline, as errors in early stages could propagate and affect overall performance.
- Author Clarification: The authors acknowledged the issue and discussed how errors were minimized but admitted that a detailed robustness analysis was lacking.
- Outcome: NS6F did not increase their score due to this unresolved concern.

(e) Presentation Issues (Reviewer H9RQ, NS6F)

- Concern: Both reviewers mentioned clarity issues, including overly lengthy descriptions and formatting problems in figures and tables.
- Author Clarification: The authors streamlined descriptions and fixed formatting issues in their revisions.
- Outcome: Both reviewers acknowledged the improvements, and H9RQ increased their score.

Summary of Reviewer Scores

- Reviewer wyLP: Raised their score after seeing revisions addressing specific concerns about the impact of knowledge bases and the inconclusive rate.
- Reviewer H9RQ: Increased their score due to clarifications on the novelty of off-target hallucinations and additional experimental results.
- Reviewer NS6F: Maintained their score, citing unresolved concerns about computational overhead, cascading errors, and broader dataset evaluations.

Overall, the reviewers acknowledged the authors’ efforts to address feedback, but significant concerns about generalizability, practicality, and robustness persisted.

---

### Decision · Program_Chairs · 2025-01-22

Reject